# KwikBucks: Correlation Clustering with Cheap-Weak and Expensive-Strong Signals

**Sandeep Silwal**[†1], **Sara Ahmadian**[2], **Andrew Nystrom**[2], **Andrew McCallum**[2],
**Deepak Ramachandran**[*2], **Mehran Kazemi**[*2]
[1] MIT, [2] Google Research
silwal@mit.edu, {sahmadian, nystrom, mccallum,
ramachandrand, mehrankazemi}@google.com

## Abstract

The unprecedented rate at which machine learning (ML) models are growing in size necessitates novel approaches to enable efficient and scalable solutions. We contribute to this line of work by studying a novel version of the Budgeted Correlation Clustering problem (BCC) where along with a limited number of queries to an expensive oracle for node similarities (e.g. a large ML model), we have unlimited access to a cheaper but less accurate second oracle. Our formulation is inspired by many practical scenarios where coarse approximations of the expensive similarity metric can be efficiently obtained via weaker models. We develop a theoretically motivated algorithm that leverages the cheap oracle to judiciously query the strong oracle while maintaining high clustering quality. We empirically demonstrate gains in query minimization and clustering metrics on a variety of datasets with diverse strong and cheap oracles. Most notably, we demonstrate a practical application in text clustering based on expensive cross-attention language models by showing that cheaper (but weaker) embedding-based models can be leveraged to substantially reduce the number of inference calls to the former.

## 1 Introduction

Modern ML techniques have made incredible advances at the cost of needing resource-intensive models (Sharir et al., 2020). Many recent approaches are so resource-intensive that despite amazing accuracy, they only serve as proof-of-concepts and are infeasible to be scaled as-is in practical usage. The total effect of all such deployments on energy usage is also a major sustainability concern (Wu et al., 2022). While the high performance of these models motivates incorporation of their signal, their high inference cost limits the interactions that any practical algorithm can have with them.

With the increased cost in querying ML models, the cost of obtaining similarities between objects of different types (texts, images, etc.) has also substantially increased. In this paper, we aim to answer a challenging question when working with such costly similarity measure models: how can we group similar objects together when similarities of objects are obtained via expensive queries? This problem can be naturally cast as a popular and versatile clustering framework, named *Correlation Clustering (CC)*, which has been extensively studied over the past 15+ years (Bonchi et al., 2022): given similarities between arbitrary objects represented as a graph, CC minimizes a natural objective that attempts to cluster together similar vertices while simultaneously separating dissimilar ones. The high cost of querying large ML models motivates the use of the Budgeted CC (BCC) setting studied in (Bressan et al., 2019; García-Soriano et al., 2020b) where relationships between nodes are determined by making a limited number of queries to an oracle, e.g. a large ML model.

We posit that in many practical settings, coarse but efficient approximations of an expensive model can be obtained through substantially cheaper but weaker models. These weaker models can be used as a guide to spend the query budget for the expensive model more carefully. A motivating example, which heavily inspires our work, is in text clustering where one wishes to obtain similarity signals from the latest highly-accurate *cross-attention (CA)* language models (e.g., (Brown et al.,

---

*Co-advised. † Work done while interning at Google Research.

2020; Thoppilan et al., 2022)), but may be hindered by the computational burden as obtaining each pair-wise similarity between data points requires an inference call to the model, giving rise to a worse case $O(n^2)$ inference calls, where $n$ is the number of data points. *Embedding based models* (e.g., (Mikolov et al., 2013; Devlin et al., 2018) can come to the rescue as they require only $O(n)$ inference calls to obtain embedding vectors for each data point that can then be used for fast similarity computation. While embedding models typically produce substantially lower quality similarity signals than CA models (see, e.g., Menon et al. (2022)), they can still provide a good approximation to guide where the budget for the CA model should be spent.

Inspired by the above, we introduce a variant of BCC where, along with a limited number of queries to an expensive oracle, we also have unlimited access to a cheaper but less accurate second oracle. This variant of BCC bridges algorithm design and practical considerations. Indeed, a recent book (Bonchi et al., 2022) on CC states "*A further intriguing question is whether one can devise other graph-querying models that allow for improved theoretical results while being reasonable from a practical viewpoint.*" This is exactly the gap our work fills through the introduction of a query-efficient setting with access to two oracles with differing quality and cost.

We develop an algorithm dubbed `KwikBucks` that extends the well-known **Kwik**Cluster algorithm to **bu**dgeted CC with **c**heap-wea**k** and expensive-**s**trong signals. `KwikBucks` uses the weak signal as a guide to minimize the number of calls to the strong signal. Under the assumption that the weak signal returns a strict superset of the strong signal edges, our algorithm can approximately match the performance of `KwikCluster`, i.e., a 3-approximation, using only an exceedingly small fraction of all possible queries to the expensive model (Theorem 2.1). In our experiments, we strengthen our theoretical modelling with several well-motivated optimizations and demonstrate that `KwikBucks` manages to produce high quality clusterings with only a small number of queries to the expensive oracle even when there is only a weak correlation between the weak and strong signal.

We conduct extensive experiments with multiple well-studied datasets to evaluate the performance of `KwikBucks` over natural extensions of previous algorithms for closely-related problems. In all settings, `KwikBucks` recovers the best clustering solution with a much smaller strong signal budget than the alternatives, and in many cases it finds asymptotically better solutions as well. Our algorithm is also robust to the choice of weak signal oracle across different dataset settings and obtains significant improvements over five baselines — **64%** relative improvement in clustering quality (measured in terms of F1 score) when averaging over 9 datasets, and over $>$ **3.5x** reduction in query complexity compared to the best baseline.

Lastly, Our contributions can be summarized as follows:

- Introducing a novel formulation of the BCC problem which strengthens the connection between theory and practice through the interplay between algorithm design and modern ML, where a cheap-weak signal guides the queries made to the expensive-strong signal,

- Developing an algorithm for this setting with strong theoretical motivations/guarantees,

- Identifying a highly impactful application domain (text clustering) where the introduced formulation and the developed algorithm are effectively applicable,

- A comprehensive empirical analysis showing large gains over extensions of existing algorithms for closely-related problems.

## 1.1 RELATED WORK

Our paper spans correlation clustering, clustering with budget constraints, and learning from multiple annotators. For brevity, we focus on the closely related key works in these areas.

**Correlation clustering** is one of the most well studied graph clustering problems and has been actively researched over the past 15+ years (see the book (Bonchi et al., 2022)). It has numerous applications in ML and beyond, including spam detection (Ramachandran et al., 2007; Bonchi et al., 2014), social network analysis (Bonchi et al., 2015; Tang et al., 2016), entity resolution (Getoor & Machanavajjhala, 2012), and many others (Gionis et al., 2005; Hassanzadeh et al., 2009; Cohen & Richman, 2002; Kim et al., 2011). Bansal et al. (2004) introduced and gave the first constant factor approximation for complete graphs (see Def. 1.1). Variants include incomplete signed graph (Bansal et al., 2004; Ailon et al., 2008), where the problem is APX-Hard (Demaine et al., 2006), and weighted graphs (Charikar et al., 2005), where it is Unique-Games hard (Chawla et al., 2006).

**Clustering under budget constraints** studies the problem of a limited number of pairwise similarity queries. In this setting, a line of work looked at spectral clustering on partially sampled matrices: Fetaya et al. (2015) in general setting, and Shamir & Tishby (2011) and Wauthier et al. (2012) for bi-partitioning. The most relevant works to our paper are those of García-Soriano et al. (2020b) and Bressan et al. (2021) who devised algorithms for correlation clustering that given a budget of $Q$ queries attain a solution whose expected number of disagreements is at most $3\cdot \text{OPT} + O(\frac{n^3}{Q})$, where OPT is the optimal cost for the instance. Another closely related line of work studies "same-cluster" queries for various clustering problems including CC (Ailon et al., 2018; Saha & Subramanian, 2019). The differences between these works and ours are (1) they assume *all* $\binom{n}{2}$ similarity queries are already known in advance whereas we must query the strong signal to obtain similarities, (2) their queries give access to the *optimal* clustering, whereas we only query for edge signs.

**Learning from multiple annotators** considers cost-effective learning from multiple annotators where the cost of a labeler is proportional to its overall quality. The most relevant work to our setting is (Guha et al., 2015) as it considers hierarchical clustering which uses lightweight similarity scores to identify candidate pairs with high similarity and they present a near-linear algorithm that is 8 to 30 times faster than previous algorithms in practice; see Section $A$ for a detailed comparison. Lastly we survey additional related works on learning from multiple annotators as well as algorithms with predictions in Section A.

## 1.2 PRELIMINARIES AND NOTATION

The input of correlation clustering is a complete undirected graph $G = (V, E^+ \cup E^-)$ on $|V| = n$ vertices. $E^+$ and $E^-$ represent the partitions of all possible $\binom{n}{2}$ edges where an edge $e = (u, v) \in E^+$ indicates that $u$ and $v$ are similar and $e = (u, v) \in E^-$ indicates that $u$ and $v$ are dissimilar. We simplify the notation to $G = (V, E = E^+)$ so any present edge is a positive edge and any missing edge is a negative edge. Additionally, we use $m$ to denote the size of $|E|$ and $\Gamma(v) = \{u \mid (v, u) \in E\}$ to denote the neighborhood of vertex $v$.

A *clustering* is a partitioning $C = \{C_1, C_2, \cdots\}$ of $V$ into disjoint subsets. Let $C_{v,u}$ denote the indicator variable if the vertices $v$ and $u$ are assigned to the same cluster. We study the min-disagreement formulation of the correlation clustering problem defined as follows (Bansal et al., 2004).

**Definition 1.1** (Correlation Clustering (CC)). *Given a graph $G = (V, E)$, the objective of correlation clustering (CC) is to output a clustering $C$ that minimizes:*

$$\sum_{e=(v,u)\notin E} C_{v,u} + \sum_{e=(v,u)\in E} (1 - C_{v,u}). \tag{1}$$

The most well-known CC algorithm is `KwikCluster` (Ailon et al., 2008), which proceeds by successively picking a vertex $p$, called a pivot, uniformly at random from the graph and forming a cluster $\{p\} \cup \Gamma(p)$. The algorithm removes this cluster and recurses on the remaining graph until all vertices are assigned to clusters. Based on the fact that the set of pivots is a maximal *independent set* constructed from a random order of vertices, Bonchi et al. (2013) suggests an equivalent algorithm that first constructs the independent set and then assigns any non-pivot to its first neighbor in the independent set. Both algorithms yield 3-approximation in expectation, however the second algorithm is more efficient as the assignment of non-pivots can be performed in parallel.

Despite practicality and simplicity of `KwikCluster` (and its variants), the algorithm assumes access to the full similarity graph $G$ and is not feasible when similarity measures are expensive to acquire. We consider budget CC studied before by (García-Soriano et al., 2020a; Bressan et al., 2019) where there is a limit (budget) for the number of queries that can be made.

**Definition 1.2** (Expensive / Strong Oracle). *Given an edge $e$, the query $\mathcal{O}_S(e)$ outputs whether $e \in E$, i.e., $e$ is a positive edge.*

Following the motivations provided in Section 1, we also introduce a second weaker oracle which is cheaper to query.

**Definition 1.3** (Cheap / Weak Oracle). *Given any vertex $v$, the query $\mathcal{O}_W(v)$ outputs a similarity score in $\mathbb{R}$ between $v$ and every other vertex in $V$, where higher values indicate higher similarity*

We frequently refer to the graph $G$ as the *strong signal graph* and likewise a strong signal edge refers to an edge in $E$. We interchangeably use the terms signal or oracle, the terms strong and expensive signal, and also the terms weak and cheap signal.

## 2 THEORETICAL MODELLING FOR ALGORITHM DESIGN

We introduce an algorithm that leverages the cheap signal for strong signal query efficiency. Our goals are twofold: **(1)** Design a flexible algorithm paradigm which can adapt to incorporate constraints necessitated by practice, i.e., limited access to expensive queries, **(2)** Analyze the quality of the produced solution with respect to the CC objective (see equation 1). We first introduce a modelling assumption for the weak oracle for the purpose of theoretical analysis. While this results in a *different* but related weak oracle formulation compared to Definition 1.3, it lets us derive a robust algorithm design which we subsequently adapt to the more realistic setting of Definition 1.3.

First, we introduce a noise factor $\gamma$ that determines the usefulness of a weak signal. $\gamma = 0$ corresponds to a perfect weak signal that exactly matches the strong signal and $\gamma = n$ corresponds to a completely uninformative weak signal.

**Assumption 2.1.** *For a fixed noise parameter $\gamma > 0$, the query $\mathcal{O}_W^\gamma(v)$ outputs a subset of $V$ such that $\Gamma(v) \subseteq \mathcal{O}_W^\gamma(v)$ and $|\mathcal{O}_W^\gamma(v)| \leq (1+\gamma)|\Gamma(v)|$.*

The existence of such a signal with a small $\gamma$, say $O(1/n)$ or $\gamma < 1$ might seem like a strong assumption for most applications. However, our experiments show that weak signal can actually provide predictive hints about the true underlying strong signal graph. More precisely, given vertex $v$, we order $V$ with respect to the weak signal, and observe that *true* strong signal neighbors of $v$ are often ranked higher. Thus the simple procedure of returning the most similar vertices for an input node captures many of the true strong signal neighbors of $v$ and mimics the clean abstraction of Assumption 2.1 (See Appendix F.6 for further empirical justification).

Using the above characterization, we next explain the high level ideas of our algorithm `KwikBucks` (Algorithm 1). It is inspired by a variant of `KwikCluster` (Bonchi et al., 2013) adapted to our two oracle setting. In this variant, we first pick pivots by forming a maximal independent set from a random ordering of vertices, and then assign non-pivots to their first neighbor (which must exist by maximality of the independent set). A naive extension of this algorithm can result in $\Omega(n^{1.5})$ queries to the strong signal (see B.1). However, one may argue that by using the weak signal, we can prune the possible neighborhood of vertices which results in fewer strong signal queries.

While the weak signal can help us make smarter queries to the strong signal, we can still show that even for a weak signal with a small error rate, i.e., $\gamma = O(1)$, we still need $\Omega(n^2)$ queries to the strong signal when forming a maximal independent set (see B.2). To circumvent this difficulty, we consider another modification of `KwikCluster` by Bonchi et al. (2013) where instead of picking a maximal independent set, we pick $t$ vertices uniformly at random and then pick an independent set from them. The caveat of this approach is that some non-pivots may not have any neighbor in the chosen pivot set, and so these non-pivots are returned as singleton clusters. This results in an algorithm which returns a solution with cost at most $3OPT + O(n^2/t)$. We additionally modify this algorithm by incorporating the weak signal to further prune possible strong signal queries. While this algorithm has an additive error for correlation clustering cost, it helps us direct our queries to "impactful" portions of graph. We now have all the ingredients (and motivation behind them) for describing our algorithm, `KwikBucks`.

`KwikBucks` first picks *all* pivots via random sampling as shown in `GetPivots`: each sampled vertex is added to the pivot set if it is not connected to the current subset of pivots that is trimmed down by `WeakFilter` (which uses the weak signal). Then, it continues to assign non-pivots to clusters, through `AssignToClusters`, which finds the first vertex (`FirstNeighbor`) in the subset of ordered pivots trimmed down by the weak signal via `WeakFilter`. If no such vertex exists, then the vertex is assigned to its own cluster, i.e., a singleton cluster. Note that having a small $t$ (the number of sampled vertices) helps query efficiency by functionally reducing the set of vertices that the weak signal is applied to (both when selecting pivots and when assigning to pivots) and then further queried by the strong signal. This comes at the cost of a small additive error. Our next theorem formally bounds the number of queries and the effect of $t$ (See Appendix B for full proof).

**Algorithm 1** KwikBucks (Our Algorithm)

**Require:** A bound on sampled vertices, $t$, the strong signal budget, $Q$.
1: $P \leftarrow \text{GetPivots}(t, Q)$
2: **return** $\text{AssignToClusters}(P, V \setminus P, Q)$

**Algorithm 2** AssignToClusters$(P, U, Q)$

**Require:** List of pivots, $P$, a vertex set, $U$, remaining strong signal budget, $Q$.
1: $A \leftarrow \emptyset$ {the set of singletons}
2: $C_p \leftarrow \{p\}$ {cluster for any pivot $p \in P$}
3: **while** $Q > 0$ and $U \neq \emptyset$ **do**
4:   $v \leftarrow$ extract first vertex of $U$
5:   $N_v \leftarrow \text{WeakFilter}(v, P)$
6:   $p \leftarrow \text{FirstNeighbor}(v, N_v, Q)$
7:   **if** $p \neq \emptyset$ **then**
8:     $C_p = C_p \cup \{v\}$
9:   **else**
10:     $A \leftarrow A \cup \{v\}$
11: $A \leftarrow A \cup U$
12: **return** $\cup_{p \in P} C_p \cup_{v \in A} \{v\}$

**Algorithm 3** GetPivots$(t, Q)$

**Require:** A bound on the number of sampled vertices, $t$, the remaining strong signal budget, $Q$.
1: $\{v_1, \ldots, v_t\} \leftarrow t$ sampled vertices
2: $P \leftarrow \{v_1\}$
3: **for** $i \geq 2$ **do**
4:   $N_i \leftarrow \text{WeakFilter}(v_i, P)$
5:   **if** $\text{FirstNeighbor}(v_i, N_i, Q) = \emptyset$ **then**
6:     $P \leftarrow P \cup \{v_i\}$
7: **return** $P$

**Algorithm 4** FirstNeighbor$(v, N, Q)$

**Require:** Input vertex, $v$, an ordered list of vertices, $N$, the remaining strong signal budget, $Q$.
1: **while** $Q > 0$ and $N \neq \emptyset$ **do**
2:   $u \leftarrow$ extract first vertex from $N$
3:   $Q \leftarrow Q - 1$
4:   **if** $\mathcal{O}_S(v, u) = 1$ **then**
5:     **return** $\{u\}$
6: **return** $\emptyset$

**Algorithm 5** WeakFilter$(v, S)$

1: **Return** $\mathcal{O}_W^\gamma(v) \cap S$

**Theorem 2.1.** *Under Assumption 2.1,* KwikBucks *uses* $n + t + 2\gamma t m/n + 2\gamma t^2 m/n^2$ *queries to* $\mathcal{O}_S$ *to achieve approximation* 3OPT + $O(n^2/t)$.

*Proof Sketch.* For a fixed vertex $u$, the number of vertices incorrectly identified as neighbors of $u$ by $\mathcal{O}_W(u)$ is bounded by $\gamma|\Gamma(u)|$, so FirstNeighbor algorithm makes at most $\gamma|\Gamma(u)| + 1$ queries to the strong oracle. The additional $+1$ term for sampled $t$ vertices and non-pivots can be bounded by $n + t$ term. So we need to bound $\gamma|\Gamma(u)|$ for an arbitrary vertex $u$. Now for each vertex $u$, since we look at intersection of its weak signal neighbors and pivots, on average we expect $t/n \cdot \gamma|\Gamma(u)|$. Since $\sum_{v \in V} |\Gamma(v)| = 2m$, this means that the total number of calls from non-pivots can be bounded by $2\gamma t m/n$. For randomly chosen vertices, we can again apply the similar argument with additional information that a vertex is included in this set by probability $t/n$ and hence we get $2\gamma t^2 m/n^2$. The approximation guarantees of our algorithm follow from those of KwikCluster with a caveat: it is possible that some non-pivots do not connect to any pivot. Picking a suitable value for the number of initially sampled pivots ensures that such vertices only introduce a small additive error. $\square$

Our next corollary, considers the interesting case of a constant-size pivot set, i.e., $t = 1/\epsilon$, which will incur an additive error of $\varepsilon n^2$. This can be thought as the 'right scale' as we make a mistake on only an $\varepsilon$ fraction of all edges. We complement our corollary by presenting a matching lower bound in the appendix (Lemma B.7) showing that $\Omega(n + d\gamma/\varepsilon)$ strong signal queries are necessary to obtain the guarantees of Corollary 2.2.

**Corollary 2.2.** *Let $d$ be the average degree of the strong signal graph and suppose $n$ is sufficiently large ($n > 1/\varepsilon$). We can achieve approximation* 3·OPT $+\varepsilon n^2$ *with* $n + O(d\gamma/\varepsilon)$ *queries to* $\mathcal{O}_S$.

Importantly, our design is modular, i.e. composed of a few fundamental components which can be individually optimized based on practical constraints. In the next section, we augment some of these components to address pragmatic considerations to better align the algorithm to practice.

## 3 THE FINAL EMPIRICAL ALGORITHM

We now extend the algorithm for the idealized setting of Section 2 into a practical version of KwikBucks for general weak signals, i.e. Definition 1.3. While this version does not satisfy

Assumption 2.1 and hence does not have similar approximability guarantees, it still retains some theoretical motivation (sketched below), and is empirically very successful (see Section 4).

The modifications we make for our practical algorithm are based on the following natural inductive bias: *'similar' edges according to the strong signal are likely to have a high weak signal similarity score*. At a high level, we incorporate this assumption throughout our algorithm design by ranking potential queries to the strong signal according to weak signal similarity values.

**Modifying `WeakFilter` to `WeakFilterByRanking`.** The most noticeable change occurs for `WeakFilter`: In our theoretical modelling, it returns a subset of $S$ which intersects with the noisy neighborhood returned by the cheap oracle. For the general weak signal version (Definition 1.3), we update the `WeakFilter` function to instead rank the vertices in $S$ with respect to the weak signal similarity to $v$ and then output the top $k$ elements in $S$ with the highest similarities (Algorithm 6)). Intuitively, for a suitable parameter $k$, the top $k$ candidates capture many of the strong signal neighbors of $v$ in $S$. Indeed, we empirically verify this in our experiments and show that predictive weak signals usually rank true strong signal neighbors much higher compared to a random ordering. For our experiments we fix $k = 100$ and perform ablation studies on this parameter.

To better understand the effect of this modification, consider `AssignClusters` where non-pivot vertices attempt to connect to a pivot. In our theoretical modeling and in the classical `KwikCluster` algorithm, each non-pivot vertex checks for a strong signal edge

---

**Algorithm 6** `WeakFilterByRanking`$(v, S, k)$

---

**Require:** Input vertex $v$; set $S \subseteq V$
1: $w_i \leftarrow$ similarity of $(v, u_i)$ for all $u_i \in S$ as computed by weak signal
2: Sort elements of $S$ in decreasing $w_i$ values
3: **Return** First $k$ elements of new sorted order

---

among the list of pivots in an ordering which is fixed for all vertices. This ordering can be thought of as the ordering inherited from `GetPivots`. In contrast, `WeakFilterByRanking` introduces a data adaptive ordering step where each non-pivot vertex can re-rank pivots based on weak signal similarities. As shown in Section 4, this has a sizeable impact on the empirical performance of our algorithm. In Section D.1, we explain these gains by introducing a natural data model that makes some well-motivated assumptions about the relationship between the strong and weak signal, as well as the inherent clusterability of the underlying graph. Under this model, we prove that the quality of the clustering after re-ranking is strictly better than for the unranked filter.

**Further optimizations.** Exploiting the ranking theme further, we make three additional enhancements to `KwikBucks`. The first one simply sorts the non-pivots based on the maximum weak signal similarity to the pivots so that 'easier' non-pivots are assigned clusters first which improves query efficiency. The second one modifies the `WeakFilterByRanking` function slightly by increasing the weak signal similarity value between a non-pivot $v$ and a pivot $p$ if $p$ has 'many' nearest neighbors (in weak signal similarity) of $v$ already in its cluster. Finally, the last enhancement introduces a post-processing step where we potentially merge some clusters after our algorithm terminates. As shown in Section D.2, this optimization is motivated by a theoretical worst-case example for `KwikCluster`. The merging step proceeds by first curating a list of clusters to consider for merging based on the average weak signal value between the two clusters and we sample a small number of strong signal edges between potential clusters to merge to determine if the pair is suitable for merging. Each of these optimizations is described in detail in Section C.

## 4 EXPERIMENTS

**Datasets.** We use 9 datasets, 8 publicly available and 1 proprietary internal. Each dataset exhibits different properties such as varying strong signal graph densities and diverse strong and weak signals to demonstrate the versatility of our method. We provide high-level descriptions here and refer to Section E for more details.

Four public datasets are comprised of text inputs: Stackoverflow (SOF) (Xu et al., 2017), Search-Snippets (Phan et al., 2008), Tweet (Yin & Wang, 2016) and AgNews (Rakib et al., 2020). For Stackoverflow and SearchSnippets, we use word2vec embedding similarities (Mikolov et al., 2013) as the cheap signal and a large cross-attention based language model as the strong signal. For Tweet and AgNews, BERT embedding similarities (Devlin et al., 2018) are the cheap signal; the strong

signal of an input pair is the indicator variable of the two examples belonging to the same class plus a small amount of i.i.d. noise to prevent the formation of disconnected connected components, which is the 'easy' case for `KwikCluster` [1].

The other four public datasets are comprised of attributed graphs: Cora (Sen et al., 2008), Amazon Photos (Shchur et al., 2018), Citeseer (Sen et al., 2008), and Microsoft Medicine (Shchur & Günnemann, 2019). For Cora and Amazon photos, node embedding (learned using deep graph infomax (Velickovic et al., 2019)) similarities are the cheap signal; the strong signal is generated similarly to those of Tweet and AgNews. For Citeseer and Microsoft Medicine, node attribute similarities are the cheap signal and the existence/absence of edges in the graph is the strong signal.

Moreover, we report results on a large proprietary dataset based on the shopping reviews of a commercial website. We use internally developed (and finetuned) embedding based and cross-attention based language models for the cheap and expensive signals respectively; both models are based on the publicly available language models such as BERT and T5 (Raffel et al., 2020).

**Baselines.** Since our work is the first correlation clustering algorithm which utilizes both strong and weak signals, we adapt algorithms from prior work, e.g. some which only use a strong signal, to our setting. In addition we propose several new natural algorithms as baselines.

- **Baseline 1:** A variant of `KwikBucks` where we do not use the weak signal ordering computed in Algorithm 6 when checking for a strong signal edge between a node and a set of pivots. Rather we use the the order the pivots were picked (each vertex always queries pivots in the same order). This is the ordering necessary for the theoretical guarantees of the `KwikCluster` algorithm and our guarantees of Section 2.
- **Baseline 2:** Algorithm presented in (García-Soriano et al., 2020a; Bressan et al., 2019). It follows the `KwikCluster` algorithm and uses the strong signal to query edges. If the query budget is depleted, the algorithm is terminated and any remaining vertices are returned as singletons.
- **Baseline 3 / 4:** We compute a $k$-NN graph based on the weak signal to narrow down the set of all possible queries to a small set of relevant pairs. Each edge of the $k$-NN graph is re-weighted (either $0$ or $1$) based on the strong signal. Baseline 3 runs the classic spectral clustering algorithm and baseline 4 runs the vanilla `KwikCluster` algorithm to completion on this graph.
- **Baseline 5:** This baseline is inspired by the baseline used in (García-Soriano et al., 2020a). We pick $k$ random vertices and query their complete neighborhood using the strong signal. $k$ is again chosen as high as possible within the allotted query budget. Instead of running an affinity propagation algorithm, which was already shown in (García-Soriano et al., 2020a) to be inferior to Baseline 2, we simply run the vanilla `KwikCluster` algorithm to completion on this graph.

**Evaluation metrics.** We evaluate our algorithm and baselines based on the correlation clustering objective (equation 1). For the purpose of evaluating metrics, we use all edges of the strong signal graph in contrast to the duration of algorithm execution which we limit the access. In addition, we compute the **precision** and **recall** of edges of the strong signal graph. Given a clustering $\mathcal{C}$, its precision is defined as the ratio between the number of strong signal edges whose endpoints are together in the same cluster and the total number of pairs of vertices clustered together in $\mathcal{C}$. The recall is defined as the fraction of all strong signal edges whose vertices are clustered together in some cluster of $\mathcal{C}$; see equation 2 and 3.

These values represent two extremes which must be balanced: recall is maximized if all vertices are clustered together whereas precision is maximized if clusters just consist of single edges. These values are more meaningful than the CC objective alone in cases where the strong signal graph is sparse (see Section F). It's not always possible to maximize both precision and recall; Section D.3 provably shows an inherent trade-off between these values. We combine the precision and recall into a single metric via the standard $F_1$ score and report this value as a function of the query budget. In our experiments, we plot averages of 25 trials and the standard deviation.

**Parameter configurations.** Our algorithm has two main parameters to select: $t$ in Algorithm 3 corresponding to the number of vertices we select uniformly at random which is then pruned to form the set of pivots, and $k$ in Algorithm 6 corresponding to the number of top vertices we select based on the weak-signal similarity for the strong signal to query. We pick both these parameters in a data-driven manner. Thorough motivation and trade-offs associated with both parameters are

---

[1]One can easily show that in such a case the classical `KwikCluster` algorithm is able to recover OPT.

Table 1: $F_1$ values for a fixed budget of $3n$ to the expensive-strong signal, where $n$ indicates the dataset size. For Citeseer and Medicine, we use a budget of $50n$ as they have substantially sparse graphs. Winners are in bold and second winners are underlined.

| | SOF | Search | Tweet | AgNews | Cora | Photos | Citeseer | Medicine | Internal | Avg. |
|---|---|---|---|---|---|---|---|---|---|---|
| B1 | $.13_{\pm.02}$ | $.73_{\pm.15}$ | $.02_{\pm.00}$ | $\underline{.74}_{\pm.01}$ | $.57_{\pm.08}$ | $.44_{\pm.01}$ | $.07_{\pm.01}$ | $.02_{\pm.00}$ | $.00_{\pm.00}$ | .30 |
| B2 | $.28_{\pm.10}$ | $\underline{.81}_{\pm.12}$ | $.15_{\pm.07}$ | $\underline{.74}_{\pm.01}$ | $\underline{.58}_{\pm.13}$ | $.53_{\pm.13}$ | $.09_{\pm.01}$ | $.03_{\pm.01}$ | $\underline{.05}_{\pm.04}$ | $\underline{.36}$ |
| B3 | $\underline{.33}_{\pm.07}$ | $.70_{\pm.04}$ | $\underline{.21}_{\pm.03}$ | $.66_{\pm.04}$ | $.54_{\pm.02}$ | $\underline{.66}_{\pm.05}$ | $.00_{\pm.00}$ | $.00_{\pm.00}$ | - | - |
| B4 | $.01_{\pm.00}$ | $.01_{\pm.00}$ | $.03_{\pm.00}$ | $.00_{\pm.00}$ | $.01_{\pm.00}$ | $.00_{\pm.00}$ | $\mathbf{.46}_{\pm.01}$ | $\underline{.25}_{\pm.00}$ | $.00_{\pm.00}$ | .08 |
| B5 | $.00_{\pm.00}$ | $.00_{\pm.00}$ | $.00_{\pm.00}$ | $.00_{\pm.00}$ | $.00_{\pm.00}$ | $.00_{\pm.00}$ | $.04_{\pm.01}$ | $.00_{\pm.00}$ | $.00_{\pm.00}$ | .00 |
| **KwikBucks** | $\mathbf{.72}_{\pm.05}$ | $\mathbf{.92}_{\pm.05}$ | $\mathbf{.28}_{\pm.04}$ | $\mathbf{.87}_{\pm.00}$ | $\mathbf{.82}_{\pm.02}$ | $\mathbf{.83}_{\pm.00}$ | $\underline{.41}_{\pm.01}$ | $\mathbf{.29}_{\pm.00}$ | $\mathbf{.14}_{\pm.01}$ | **.59** |

presented in Section F.2; ablation studies of these parameters are provided in our empirical results. Lastly, we always reserve $10\%$ of the query budget for performing the merge post processing step. If the main algorithm terminates with remaining budget, we correspondingly increase the merge post processing budget to incorporate this.

**Results.** We highlight the key experimental themes and defer additional details to Appendix F.

**Superior performance over baselines:** Table 1 shows that our algorithm outperforms the baselines in terms of the $F_1$ metric: it consistently has the highest $F_1$ value for the fixed query budget result displayed in Table 1. For example for the SOF dataset, the best baseline has a **2.2x** factor smaller $F_1$ value. Figures 1(a),(b) show the CC objective and $F_1$ score as a function of the query budget for the SOF dataset. It shows that our algorithm achieves a higher $F_1$ score and a lower correlation clustering objective value with only $\approx 7 \cdot 10^3$ queries whereas the baselines require at least $25 \cdot 10^3$ queries to match KwikBucks with $7 \cdot 10^3$ queries, showing the efficacy of our algorithm with a **3.6x** reduction in query complexity. Intuitively, the weak signal allows us to make clustering progress much faster by directing the query budget to impactful strong signal queries after filtering using the weak signal. The results for other datasets are deferred to Figures 3 and 4 in the appendix which display qualitatively similar behaviour. The strong signal graphs of Citeseer and Medicine are quite sparse. Therefore for these datasets, the trivial clustering of all singletons already achieves a very low CC objective score. As argued above, in these cases the $F_1$ score is a much more meaningful measure of cluster quality. As shown in Figures 4, our algorithm achieves superior $F_1$ values compared to the baselines. Lastly we note that the performance of our algorithm stabilizes once it has exploited sufficiently many strong signal queries. We note that B3 is omitted from the CC objective value plots for clarity as it always had much higher objective value than other algorithms.

**Relative performance of baselines is dataset dependent:** As shown in Table 1, for many datasets such as Cora, Search, and AgNews, B2 is the best among our five baselines. However this does not generalize across all datasets. As shown in Figures 4, B3 is the best baseline (with respect to the $F_1$ score) for the Tweet and Photos datasets while B4 is the best baseline for the Citeseer and Medicine datasets. B4 can be a competitive baseline in the case where the strong signal graph is extremely sparse, such as in Citeseer (see Figure 4). This is because the weak signal $k$-NN graph is able to recover many relevant edges of the (sparse) graph if the weak signal is informative.

**Varying weak signal performance:** We perform addition weak signal ablation studies with the SOF and Search datasets. We replace the Word2Vec (W2V) embeddings used in our cheap oracle with tf-idf embeddings and fix all other components of the algorithm. Figure 1(c) and 7 show the performance of our algorithm on these datasets and in both cases, the algorithm's performance noticeably worsens. The intuitive answer for *why* this is the case is because the alternative weak similarities computed from tf-idf embeddings are worse than W2V embeddings at ranking strong signal neighbors. We empirically verify this claim. For every vertex $v$ in the SOF dataset, we rank all other vertices in decreasing weak signal similarities to $v$. The average rank of the true strong signal neighbors of $v$ is then computed and this value is plotted in a histogram for all vertices $v$ in Figure 1(d). A 'good' weak signal should rank actual strong signal neighbors much higher than non strong signal neighbors. Indeed we observe this to be the case for the W2V embeddings and this fact is qualitatively captured the aforementioned figures which show that W2V has superior F1 score plots. We also observe that even the weaker tf-idf embeddings still provide significant gains over not using a weak signal. Overall, these experiments along with Baseline B1 empirically verify that (1) the quality of the weak signal correlates with the performance of the algorithm, and (2) the

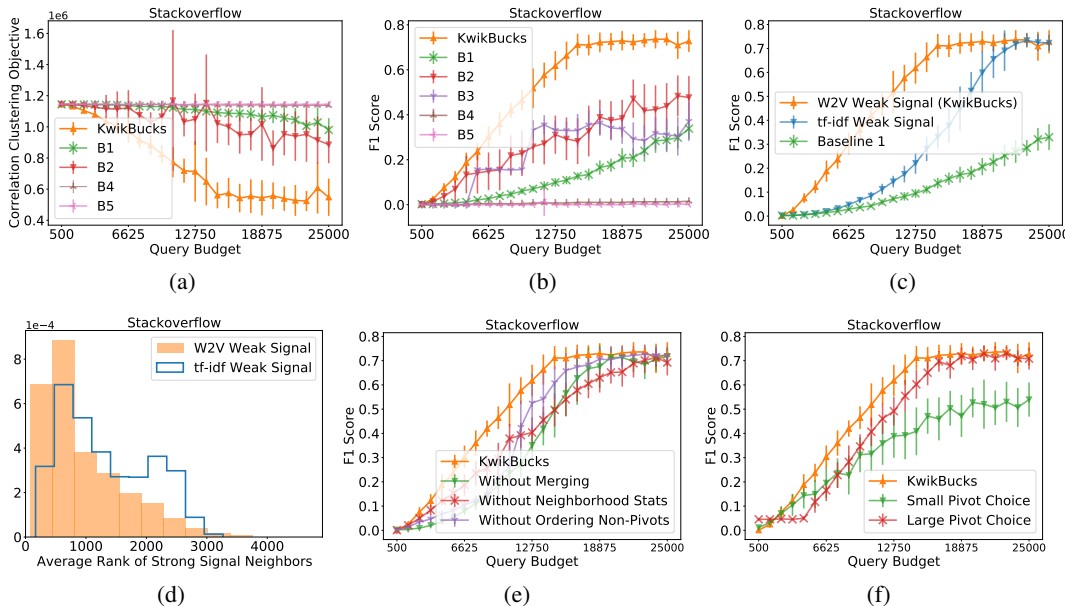

Figure 1: (a) and (b) represent the CC objective and the F1-scores for the Stackoverflow dataset across various query budgets. (c) compares performance across weak signals of various strength (Baseline 1 corresponds to a random weak signal). (d) represents a (normalized) histogram showing average rank assigned to actual strong signal neighbors by two different weak signals (lower is better). (e) represents an ablation study on some of the main components of the algorithm. (f) represents a sensitivity analysis to the parameter corresponding to the number of pivots selected.

two-oracle framework we introduced is superior than the previously studied single-oracle setting even when the cheap signal is considerably weak.

**Ablation studies.** We perform ablation studies on all tune-able parameters of our algorithm. A sample of the ablation studies for SOF is shown in Figure 1(e) and details of other results are presented in Section F.4. We observe that removing any of the main components of the algorithm (merging, ordering with respect to weak signal, and ordering with respect to the statistics of the neighboring nodes) deteriorates the performance of the algorithm, thus all the introduced components are paramount in `KwikBucks`. We also verify the role of the parameter $t$ corresponding to the number of pivots we select for our algorithm in Figure 1(f). We observe that both large and small choices for this parameters can be harmful, but choosing larger values is a safer option compared to smaller values as it asymptotically offers a similar performance as the optimal value. Section F.2 provides more details on the choice of this parameters and the other parameters of `KwikBucks`.

## 5 CONCLUSION

We introduced and studied a novel variant of the (budgeted) correlation clustering algorithm where besides having a limited query budget to an expensive-strong oracle, one also has access to a readily available cheap-weak oracle. We developed an algorithm for this setting with strong theoretical motivations. We also demonstrated strong practical motivations by showing how the proposed framework and algorithm can be effectively used for realistic clustering applications, where massively scaled ML models yield highly accurate signals but cannot be used indiscriminately. We anticipate the proposed framework could become a standard building block, especially for text clustering strategies, and envision extensions of the framework to several other ML problems where combinations of expensive-strong and cheap-weak signals are available. For example in commercial recommendation systems, the "standard practice for machine learning-driven recommendations in industry" (Wong, 2022) is a two-step procedure of cheaply retrieving a set of candidates and then iterating over them with a more powerful but costlier scoring model (Wong, 2022).

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

## A    ADDITIONAL RELATED WORKS

In the realm of learning from multiple annotators, there is a long line of work studying these both empirically and theoretically. Empirical work on this can be divided into two main streams: (1) each labeler is coming from a different generative model, (2) each labeler is an expert over an unknown subset of categories, (3) different labelers with quality proportional to their cost. In the first case, the learning algorithm focuses on learning parameters of each labeler and then for each example decides which labeler to query (Yan et al., 2011; 2012; Lin et al., 2015; 2014; Fang et al., 2012). In the second case, it uses data to measure the class-wise expertise in order to optimally place label queries (Ipeirotis et al., 2014; Donmez, 2008). In the last case, empirical results comparing designed algorithms to baselines are developed: active learning from noisy data streams (Younesian et al., 2020), active learning using diverse labelers (Huang et al., 2017), and content segmentation for personal assistants (Guha et al., 2015). Theoretical work looked at the setting where the weak labeler made mistakes mostly in heterogeneous regions of space, i.e., correct in label-homogeneous regions but may deteriorate near classification boundaries. Different formulations were considered in this setting: non-parametric setting (Urner et al., 2012), fitting classifiers in a hypothesis class (Zhang & Chaudhuri, 2015), online selective sampling with applications in linear classifiers and robust regression (Malago et al., 2014; Dekel et al., 2012).

The idea of judiciously utilizing an expensive but accurate strong model with the help of cheaper but noisier methods have already been successfully used in many practical and important domains. In nearest neighbor search and information retrieval, the dominant algorithmic paradigm is to return multiple possible nearest neighbors using scalable methods and then re-rank the returned points using exact distance calculations (which is prohibitive to perform over the entire input)[2]. In recommendation systems, the "standard practice for machine learning-driven recommendations in industry" (Wong, 2022) is driven by the two-step procedure of cheaply retrieving a set of possible candidates and iterating over them using a more powerful but costlier ML models (Wong, 2022; Bergum, 2022; Eder, 2022; goo, 2022; pap, 2022). Similar ideas are also used in question answering and vision applications (Zhong et al., 2017; Barz & Sonntag, 2021).

There has also been extensive work in incorporating additional predictions in algorithmic design for online algorithms (Bamas et al., 2020b; Purohit et al., 2018; Lykouris & Vassilvitskii, 2018; Purohit et al., 2018; Gollapudi & Panigrahi, 2019), sublinear space and time algorithms Chen et al. (2022); Hsu et al. (2019); Eden et al. (2021), and other algorithmic and data structural problems (Mitzenmacher, 2018; Bamas et al., 2020b;a; Wei & Zhang, 2020; Jiang et al., 2020; Diakonikolas et al., 2021; Charikar et al., 2001; Antoniadis et al., 2020a;b; Anand et al., 2022; Nguyen & Dürr, 2021). We refer to the Algorithms-with-Predictions website[3] for comprehensive references. The high level motivations of these works is to apply predictions to aid in beyond worst-case analysis of algorithmic problems. The prototypical examples of predictions used in these works include algorithm parameter settings (for example 'warm starts' or 'seeds' which can be constructed from past inputs). Thus a common underlying assumption is that many similar inputs are given so that predictions are meaningful and feasible. Furthermore in many of these works, the predictions are modeled after particular problem settings in mind and the inputs are always fully specified. In contrast, our predictions are inspired by a particular application domain, e.g. text clustering, which we connect to CC, rather than motivating the predictions from a purely algorithmic problem perspective. Furthermore, our predictions (e.g. queries from the weak signal) help us learn about the true underlying input (e.g. the strong signal graph).

We also give a detailed comparision to the work of (Guha et al., 2015). While at a high level both (Guha et al., 2015) and our work aggregate information across various signals, the two works differ in terms of the generality of oracles considered, the formal guarantees given, and the problems studied. The oracles used in (Guha et al., 2015) are highly specialized to the datasets at hand; for example, the cheap oracle used in (Guha et al., 2015) is an inverted index model which heavily relies on the specifics of the datasets used. In contrast, we take a broader view of weak and strong oracles and present theoretically founded algorithms which only assume query access to the weak and strong model and not any particular model idiosyncrasies. Therefore, our algorithm has provable

---

[2]We refer to http://ann-benchmarks.com/ for a large collection of practical nearest neighbor search algorithms and Andoni et al. (2018) for a overview of theoretical works.

[3]https://algorithms-with-predictions.github.io/

guarantees on both the approximation quality and the query complexity, making it broadly applicable across different oracles. In terms of problems, we study correlation clustering while the focus of (Guha et al., 2015) is not on a clustering problem. Rather, they use hierarchical clustering as an intermediate problem to perform user modeling and do not consider any specific clustering objective functions. The strong signal queries made by our algorithm are guided through formal reasoning and they exploit the structure of the clustering problem we are studying. In (Guha et al., 2015) the weak signal is used at a more intuitive level and serves the informal role of filtering possible strong signal queries with no formal reasoning.

## B    OMITTED PROOFS OF SECTION 2

**Proposition B.1.** *There exists a strong signal graph $G$ such that the* `KwikCluster` *algorithm makes $\Omega(n^{1.5})$ strong signal queries.*

*Proof.* Consider the case where the strong signal graph consists of $\sqrt{n}$ cliques, all of size $\sqrt{n}$. In this case, every time `KwikCluster` picks a pivot, it has to examine an existence of an edge from this pivot to all the unassigned vertices. So for at least the first $\sqrt{n}/2$ times `KwikCluster` picks a pivot, it has to make at least $n/2$ calls to $O_S$ , resulting in $\Omega(n^{1.5})$ calls to $O_S$. ☐

**Proposition B.2.** *Consider a variation of* `KwikCluster`*, called* `KwikCluster`$^\gamma$*, which for a chosen pivot $p$ from uncovered vertices $V'$, only queries $V' \cap \mathcal{O}_W^\gamma(p)$ from strong signal. There exists a graph $G$ such that* `KwikCluster`$^\gamma$ *still makes $\Omega(n^2)$ strong signal queries in expectation even when $\gamma = O(1)$.*

*Proof.* Consider the following strong signal graph: the graph $G$ is comprised a fully connected clique on $0.9n$ nodes. The graph also has $0.1n$ nodes, called 'outside vertices' which all connect to the same $0.1n$ vertices in the fully connected clique but have no edges between them. Suppose that $\mathcal{O}_W^\gamma$ returns the correct strong signal neighborhood for vertices in the clique but for the outside vertices, it returns an additional $\Omega(n)$ arbitrary vertices among the outside vertices.

Now consider the simulation of `KwikCluster`$^\gamma$ on $G$. With constant probability, the first time a pivot is picked, it comes from the clique vertices which have no neighbors among the outside vertices. Condition on this event. Now the algorithm still needs to run until the outside vertices have been selected in a cluster. However, every time each such vertex is picked as a pivot, we need to check over $\Omega(n)$ erroneous vertices. Furthermore, removing an outside vertex $v$ and its neighborhood $\Gamma(v)$ does not remove any of the other outside vertices. Thus it follows that the expected number of queries made to $O_S$, while still utilizing $\mathcal{O}_W^\gamma$ in this natural variant of `KwikCluster`, is $\Omega(n^2)$ with constant probability since there are $\Omega(n)$ outside vertices, each which requires $\Omega(n)$ queries to $O_S$. Altogether, this natural algorithm can incur $\Omega(n^2)$ queries, a super linear amount. ☐

**Lemma B.3.** *Let $t$ be the parameter of* `GetPivots` *(Algorithm 3). Algorithm 3 uses $t + 2\gamma t^2 m/n^2$ queries to $O_S$.*

*Proof.* Let $T = \{v_1, \ldots, v_t\}$ denote the set of $t$ sampled vertices in line 1 of Algorithm 3. Fix a vertex $v \in T$, we show that there are $O(\gamma t|E|/n + 1)$ queries in expectation for such vertex. Let $P_i$ denote the set of pivots right before we start scanning vertex $v_i$. When we check for the neighbors of vertex $v_i$, we immediately stop if a strong signal neighbor in $P_i$ is found. Let $A(v_i) = \mathcal{O}_W^\gamma(v_i) \setminus \Gamma(v_i)$ be the vertices which $\mathcal{O}_W^\gamma$ errs on for query $v_i$ and can result in needless calls to $O_S$. The number of expected calls to the strong signal is exactly the expected number of unnecessary calls $\mathbb{E}[|A(v_i) \cap P_i|]$ plus one call that may result in the early stoppage in line 4 of Algorithm 4. So

for each $v_i$, the expected number of calls to $O_S$ can be bounded by

$$
\begin{aligned}
1 + \mathbb{E}[|A(v_i) \cap P_i|] &\leq 1 + \mathbb{E}[|A(v_i) \cap T|] \\
&= 1 + \mathbb{E}_{v_i} \mathbb{E}[|A(v_i) \cap T| \mid v_i] \\
&\leq 1 + 2\mathbb{E}_{v_i} \mathbb{E}\left[|A| \cdot \frac{t}{n} \mid v_i\right] \\
&\leq 1 + \frac{2t\gamma}{n} \mathbb{E}_{v_i}[|\Gamma(v_i)|] \\
&= 1 + \frac{2t\gamma m}{n^2}.
\end{aligned}
$$

Summing over all $t$ vertices in $T$ results in the final bound. $\qquad\square$

**Lemma B.4.** *Let $P$ be the output of* `GetPivots(t,Q)`*, then* `AssignToClusters`$(P, V\setminus P, Q)$ *makes $n + 2\gamma tm/n$ queries to $O_S$.*

*Proof.* Consider a fixed vertex $u \in V \setminus P$. We perform a similar analysis as in Lemma B.3: ideally $S = \mathcal{O}_W^\gamma(u) \cap P$ informs us the pivot which $u$ should connect to. However since the cheap oracle can be noisy, we can have many vertices in $(\mathcal{O}_W^\gamma(u) \setminus \Gamma(u)) \cap P$. The number of queries to $O_S$ is at most $|(\mathcal{O}_W^\gamma(u) \setminus \Gamma(u)) \cap P| + 1$. It remains to calculate the following expected value:

$$
\mathbb{E}[|(\mathcal{O}_W^\gamma(u) \setminus \Gamma(u)) \cap P|] \leq \gamma |\Gamma(u)| \cdot \frac{t}{n}.
$$

Thus the total expected number of queries for non-pivot vertices is

$$
\sum_{u \in V\setminus P} \left(1 + \gamma |\Gamma(u)| \cdot \frac{t}{n}\right) = |V \setminus P| + \frac{\gamma t}{n} \sum_{u \ in V\setminus P} |\Gamma(u)| \leq n + \frac{2\gamma tm}{n}.
$$

$\qquad\square$

**Theorem B.5** ((Bonchi et al., 2013)). *Let $T'$ be the maximal independent set formed by scanning randomly sampled $t$ vertices of a graph $G$. Then the expected number of edges of $G$ not incident with an element of $T'$ union the neighborhood of $T'$ in $G$ is at most $n^2/2t$.*

**Theorem 2.1.** *Under Assumption 2.1,* `KwikBucks` *uses $n + t + 2\gamma tm/n + 2\gamma t^2 m/n^2$ queries to $\mathcal{O}_S$ to achieve approximation 3OPT + $O(n^2/t)$.*

*Proof.* The bound on the number of queries to $O_S$. There are two tasks which require calls to $O_S$: forming the set of pivots in `GetPivots` and assigning non-pivot vertices to a pivot in `AssignClusters`. The expected number of queries for `GetPivots` is handled by Lemma B.3 and the expected number of queries for `AssignClusters` is handled by Lemma B.4.

We now need to bound the approximation guarantee. Consider the subgraph $G'$ of the strong signal graph which is the union of the pivots returned by `GetPivots` and their neighborhood. Theorem B.5 gives us that the number of edges not part of this subgraph is at most $O(n^2/t)$ which can be charged to the additive error incurred by our algorithm (all vertices which do not have a strong signal edge to any of the pivots are clustered as singletons). Now on this subgraph note that we are exactly mimicking the `KwikCluster` algorithm on $G'$. This is because the pivots of Get-Pivots are chosen from the same distribution as the `KwikCluster` algorithm since we ensure that all pivots chosen are not in the neighborhood of previously chosen pivots. Thus we obtain a $3\cdot$ OPT guarantee on $G'$. To obtain the final guarantee on the original strong signal graph, note that the OPT clustering of $G$ restricted to $G'$ cannot be better than the OPT correlation clustering of $G'$. The result follows from considering our additive error as well. $\qquad\square$

We now show a lower bound on the query complexity of our algorithm. First we recast the lower bound result of (García-Soriano et al., 2020a) in the language of strong and weak oracles. They show that *any* algorithm which only has access to the strong signal must make $\Omega(n^3/(\Delta c^2))$ queries to obtain a $c\cdot$ OPT $+\Delta$ correlation clustering objective guarantee. We can translate their lower bound into our setting of strong and weak oracles by essentially making the weak oracle useless through

a suitable choice of $\gamma$. The lower bound shows that for constant $\varepsilon$ and $n$ large enough, Corollary 2.2 is optimal. First we formally state the guarantees given by (García-Soriano et al., 2020a) in our language of strong and weak signals.

**Theorem B.6** ((García-Soriano et al., 2020a)). *For any $c \geq 1$ and $\Delta$ such that $8n < \Delta \leq \frac{n^2}{2048c^2}$, any algorithm finding a clustering with expected cost at most $c \cdot OPT + \Delta$ must make at least $\Omega(n^3/(\Delta c^2))$ adaptive strong signal queries.*

**Lemma B.7.** *Let $\varepsilon \geq \Omega(1/n)$ be sufficiently small. In the worst case input, any algorithm must use at least $\Omega(n + d\gamma)$ strong signal queries to obtain a $3 \cdot OPT + O(\varepsilon n^2)$ approximation to the correlation clustering objective.*

*Proof.* We recall the lower bound example of (García-Soriano et al., 2020a) (which is proved in Theorem 4.1 in (García-Soriano et al., 2020a)). Let $k = n^2/(32c\Delta)$ (note that $k < n$ by design). Their worst case strong signal graph example consists of $k$ equal sized cliques and all vertices have degree $\Theta(n/k)$. Now we consider the case where the weak oracle is completely useless and always returns the entire set of vertices on any query. This corresponds to the case where $\gamma = \Theta(k)$ (for $\gamma$ defined in Assumption 2.1). Now directly applying Theorem 4.1 of (García-Soriano et al., 2020a) gives us that any algorithm which only has access to the strong signal must make at least $\Omega(n^3/(\Delta c^2))$ queries to obtain a $c \cdot OPT + \Delta$ correlation clustering objective guarantee. The theorem follows by noting that if $\Delta = \varepsilon n^2$ then any algorithm must make $\Omega(n/\varepsilon) = \Omega(n + k \cdot n/k) = \Omega(n + d\gamma)$ queries in this worst case example, as desired. Note that the valid range of $\varepsilon$ here follows from the restriction on $\Delta$ so $\varepsilon \geq 8/n$ and cannot be larger than some fixed constant. $\square$

## C ADDITIONAL ALGORITHMIC DETAILS FOR EMPIRICAL ALGORITHM.

We provide additional details on the algorithm design of Section 3.

---

**Algorithm 7** `SortNonPivots`$(T, V \setminus T)$

---

**Require:** $T$, the set of pivots; $V \setminus T$, vertices which are not pivots
  1: **for** vertices $v \in V \setminus T$ **do**
  2:    $w_v \leftarrow$ Maximum weak signal similarity
       between $v$ and any vertex in $T$
  3: **Return** $V \setminus T$ sorted in decreasing $w_v$ values

---

**Optimizing non-pivot order.** Continuing on the theme of ranking, once we curate the pivots, we need to assign the non-pivots to a pivot. To do so, we sort the non-pivots based on 'easiness' to assign to a pivot. Hence we sort the non-pivots by the maximum weak similarity to some pivot. This has the effect of utilizing our query budget as efficiently as possible as 'easier' non-pivots are checked first. We re-rank the vertices $V \setminus P$ in `SortNonPivots` before calling `AssignClusters` in `KwikBucks`.

**Utilizing Weak Signal Neighborhood.** We can use strong signal queries that we have already made for vertices in the weak signal neighborhood to further optimize the sorting of the pivots in `WeakFilterByRanking`. The inductive bias we are using is that if many vertices in the immediate weak signal neighborhood of a vertex $v$ connect to the same pivot $p$, then the likelihood $v$ having a strong signal edge to $p$ is high. Thus to better utilize our expensive query budget, we should query $(v, p)$ earlier than later. To make the intuition more precise, we simply update the similarities to pivots computed by $v$ in `WeakFilterByRanking` to account for the inductive bias. The new similarity score $w'p$ of a pivot $p$ is equal to $w_p$, the similarity score between $v$ and $p$ computed by the weak signal, plus a term for the number of vertices in the $k$-weak signal neighborhood of $v$ that are already connected to $p$:

$$w'_p = w_p + \lambda(\# \text{ of vertices in } k\text{-weak signal neighborhood of } v \text{ that are already in } p\text{'s cluster}).$$

This has the affect of 'boosting' some pivots to a higher ranking. See Section F.2 for further details.

C.1 Details on Post-processing Merging

In this section we provide the details of our post-processing merging strategy outlined in Section 3.

Let $C_1, \ldots, C_r$ be the clusters outputted by our algorithm. First we curate a list of cluster pairs to consider for merging. Then we rank the pairs in terms of suitability for merging. Finally we enumerate over the pairs in the order computed (until we run out of any query budget) and determine if the pair should be merged. Each of the three steps is described in detail below.

1. **Curating pairs of clusters.** It is prohibitive to consider all pairs of clusters (which might be super linear if there are many clusters). We again appeal to the weak signal and construct a $k$-nn weak signal similarity graph on the vertices for some small $k$, such as $k = 20$. Then we only consider pairs of clusters which are edges in the graph. More precisely, we consider the pair $(C_i, C_j)$ for merging if there is some $v \in C_i$ and $u \in C_j$ such that $(v, u)$ is an edge in the $k$-nn graph. This narrows down the number of pairs considerably.

2. **Ranking pairs by suitability.** For each pair $(C_i, C_j)$ of clusters from the prior step, we compute the average weak signal value between vertices in $C_i$ and $C_j$ respectively. We then rank the pairs in decreasing order based on this value.

3. **Determining if a pair should be merged.** Finally, we enumerate over the pairs in the order computed previously. Suppose we are deciding if we want to merge the pair $(C_i, C_j)$. We must ensure the pair has a high number of strong signal edges (more than 0.5 fraction). To do so we simply sample a small number of random pairs of vertices (say 20), one vertex from each cluster, and estimate the fraction of these random edges which are strong signal edges.

# D Theoretically Motivating Practical Modifications of the Algorithm

In this section we provide theoretical justifications for the practical modifications of our algorithm.

## D.1 Theoretically motivating raking pivots by weak signal

In the classical `KwikCluster` algorithm and our query efficient variant in the two oracle model of Section 2, it is imperative that the pivots are selected in a random order to provide theoretical guarantees on the quality of the computed clustering. Specifically, the worst case theoretical guarantees dictate that vertices must connect to the *first* pivot in the random order which they have a strong signal edge to.

Nevertheless, in our data driven optimization of the algorithm, we choose an adaptive ordering of the pivots for each vertex where the order is based on the weak-signal similarity scores. We empirically observed that this ordering is superior to the random ordering and achieves a higher clustering quality while utilizing a $>$ **3.5x** factor or more less strong signal queries. The explanation behind this improvement is two fold:

1. **Increased query efficiency:** fewer strong signal queries are used when a vertex attempts to connect to a pivot.

2. **Maintaining cluster quality:** connecting to pivots with larger weak signal similarities are high quality pivots.

The positive effects of the first point are straightforward to explain. Indeed, making the natural assumption that higher weak signal similarities are more indicative of a strong signal edge, checking pivots in the weak signal ordering leads to less queries wasted when a vertex attempts to connect to a pivot. In addition to the empirical results of Section 4, this point is further expanded upon in Figure 7 and Section F.6.

Thus the main goal of this section is to provide an intuitive and theoretically motivated understanding of the second point. While it may not be true that re-ranking pivots according to the weak signal similarities maintains the worst case guarantees proved in Section 2, we study a natural data set model where such a re-ranking provably helps. We wish to capture our data driven observations that

pivots with larger weak signal similarities are of high quality and larger weak signal values indicate better cluster relationships.

In our experiments, the weak signal scores are mostly computed using distances between embedding vectors. If a weak signal is useful, then it must have be predicative of the strong signal values, even if the weak signal is noisy. To mimic this, we consider the following general family of data sets:

- Each vertex $v$ has an associated vector $p_v \in \mathbb{R}^d$, representing it's 'true' embedding representation.

- The weak signal values are computed according to an appropriate distance measure $d$ on the embedding vectors (for example cosine or Euclidean distances) plus a random noise term $\xi$ (expanded upon shortly). This models the setting where the weak signals are helpful but noisy signals as they only have noisy access to the 'true' representations.

- There exists a function $f : \mathbb{R}^{\geq 0} \to [0, 1]$ which gives the probability of a strong signal edge. More precisely, let $p_v$ and $p_u$ denote the embedding of vertices $v$ and $u$. Then the probability of having the strong signal edge $(v, u)$ is given by $f(d(v, u))$. This is quite a general formulation as it includes a wide array of geometric or kernel similarity graphs for appropriate choices of $f$ and $d$.

For example, if $f = \exp(-x/\sigma)$ and $d$ is the Euclidean distance, then the true strong signal graph is the Gaussian kernel similarity graph where $\sigma$ is the scale of the kernel. Intuitively, the closer $u$ and $v$ are under the metric $d$, the higher probability $f$ assigns to the edge between $u$ and $v$.

We additionally impose the following clusterability assumptions on the data set. Our goal is to capture a natural underlying cluster structure which can be accessed via strong and weak signal queries. Vertices which are part of the same underlying cluster should have higher weak signal similarity scores, even if the scores are noisy, and the strong signal edges should be highly accurate. Our model defined below satisfies these intuitive criteria. Furthermore under our natural model, there is a 'true' pivot for a vertex $v$, even though $v$ may have strong signal edges to other pivots.

Our cluster assumptions on the data set is the following.

1. The 'true' embedding vectors $p_v \in \mathbb{R}^d$ can be partitioned into $k$ clusters such that all vectors in a cluster are within distance $R$ of each other.

2. All embedding vectors in different clusters are distance at least $2R$ from each other.

3. The probability of a strong signal edge is at least $1 - p$ for distances at most $2R$ and at most $p$ for distances at least $2R$. We think of $p < 1/2$ as a small parameter close to $0$.

4. Given inputs $u, v$, the weak signal outputs $d(p_u, p_v) + \xi$ where $\xi$ is uniformly random in $[-R, R]$. Thus smaller values are interpreted as having higher weak signal similarity.

Note that we only have access to the strong and weak signal values via queries and do not know the true underlying embedding vectors $p_v$. We now argue that the above assumptions are motivated and natural.

- The assumption (1) gives a cluster structure to the data and allows us to compare the classical KwikCluster algorithm and our re-ranking modification under natural clusterability assumptions. The exact formulation we are employing is inspired by the works of (Awasthi et al., 2012; Balcan & Liang, 2012; Ashtiani et al., 2016) which study clustering under similar proximity assumptions. For example, it can be easily checked that the 'margin' property assumption of (Ashtiani et al., 2016) directly implies our assumptions (1) and (2).

- Our assumption (3) is a natural and necessary assumption on the function $f$ as it ensures the true strong signal graph captures the underlying cluster structure of the inputs. This also corresponds to picking an appropriate *scale* parameter if $f$ is a kernel function, for example picking $\sigma$ in the Gaussian kernel. A judicious choice of $\sigma$ ensures that the underlying kernel similarity graph, which corresponds to the strong signal, is able to capture the cluster structure of the data set. Thus our assumption that $p \ll 1$ ensures that the similarity graph has strong inter-cluster connectivity while having sparse connectivity across different clusters. Indeed in practice, the kernel scale parameter is often picked using the 'median' rule and thus $\sigma = \Theta(R)$ is a natural choice which ensures our choice of $p$.

- Our data set construction ensures that the strong signal is 'more powerful' than the weak signal. Indeed, the weak signal only has access to the distances between the true embedding vectors up to some additive noise as stated in assumption (4). While the exact form of the random noise is not very consequential, we stick to the uniform noise model as it as several desirable properties:

  1. Given vertices $v, u, w$ where $u$ is in $v$'s true cluster (according to the true embedding vectors) and $w$ is not, the weak signal can potentially output a smaller value on query $(v, w)$ compared to $(v, u)$. Thus the weak signal incorrectly states $w$ is more similar to $v$ than $u$ due to the additive noise. For example if $d(v, u) = R$ and $d(v, w) = 2R$, this happens with probability $1/8$. Therefore the weak signal accurately reflects our desired goal of an indicative but noisy signal.

  2. The weak signal can be modeled by fast nearest neighbor search algorithms which return noisy nearest neighbor estimates. On the other hand, we imagine the strong signal as being expensive since it needs the true distances among the embedding vectors without any additive noise.

We believe that this natural graph model we examined for our algorithm modification helps explain and predict the strong empirical performance of our method. Thus our goal is to show that under the above data set modelling, re-ranking pivots based on weak signal similarity values provably helps. Assume that we have picked a pivot $u$ from each of the $k$ clusters of Assumption (1). We permute them randomly to form an ordering $u_1, \ldots, u_k$. This corresponds to the random ordering used by the KwikCluster algorithm and our theoretical algorithm of Section 2. Each non-pivot vertex $v$ re-ranks the pivots forming the ordering $u_{\pi_v(1)}, \ldots, u_{\pi_v(k)}$ where $\pi_v$ is a permutation depending on the weak signal similarities from the pivots to $v$. The weak signal similarities are calculated as detailed above: the weak signal outputs 'noisy' distances based on the true embedding vectors and smaller distances correspond to higher similarities. Note that each non-pivot vertex $v$ has a 'true' pivot $u$ corresponding to the pivot chosen from the cluster that the true embedding vector $p_v$ is part of. We say that a non-pivot vertex $v$ is correctly assigned by a clustering algorithm $\mathcal{C}$ if $\mathcal{C}$ assigns $v$ to its 'true' pivot. The following lemma shows that assigning vertices to pivots based on weak signal similarities strictly outperforms using a random order.

Intuitively, if another pivot $u'$ is ranked higher than $u$ in the random ordering, our proposed medication of re-ranking asked on weak signal similarities is likely to *correct* the ordering re-ranking $u$ to ahead. The lemma below provides theoretical justification of why this is sound and complements our experimental evaluation which demonstrates the empirical advantage of our re-ranking procedure.

**Lemma D.1.** *Consider the setting above. Let $\mathcal{C}$ be the clustering where every non pivot vertex picks the* first *pivot in this ordering that it has a strong signal edge to. Let $\mathcal{C}'$ be the clustering where each non pivot vertex re-ranks the pivots using weak signal similarities then picks the first pivot that it has a strong signal edge to. Let $A$ be the number of non-pivot vertices that $\mathcal{C}$ correctly assigns and similarly define $B$. We have $\mathbb{E}[A] < \mathbb{E}[B]$.*

*Proof.* Fix a non-pivot vertex $v$. Let $X$ denote the indicator variable for $\mathcal{C}$ correctly assigning $v$ and define $Y$ similarly for $\mathcal{C}'$. It suffices to show that $\mathbb{E}[X] < \mathbb{E}[Y]$. The lemma then follows by linearity of expectations and summing across all non-pivot vertices $v$. Note here that the expectation of each variable is with respect to the randomness used by the respective algorithms.

Let $u$ denote the true pivot of $v$. If $v$ does not have a strong signal edge to $u$ (according to the strong signal) then both algorithms will fail. Similarly, if $v$ only has a strong signal edge to $u$ and to no other pivots, then the performance of either clustering is the same. Now consider the case that $v$ has at least two strong signal edges to pivots, one to $u$ and rest arbitrary. Then the probability that $v$ is correctly classified by the random ordering is at most $1/2$. This is because if there is at least $1$ other pivot that $v$ as a strong signal edge to, then the probability that the random ordering places $u$ ahead of it is at most $1/2$.

One the other hand, the probability that $v$ is correctly classified by the weak signal ordering is strictly larger than $1/2$. To see this, we calculate the probability that $u$ has the highest weak signal similarity. Identically, it suffices to calculate the probability that the weak signal outputs the smallest noisy distance value for $u$. Recall the modelling assumptions of the data set: we know that $d(v, u) \leq R$ whereas $d(v, u') \geq 2R$ for all other pivots $u'$ that are not equal to $u$. The weak signal outputs

$d(v, u')$ plus a uniformly random value in $[-R, R]$. Let $\xi_u$ be the random value added for $d(v, u)$. With probability $1/2$, this value is negative so the noisy distance computed by the weak signal is strictly smaller for $u$ than all other pivots $u'$ (since in the best case, their distance is at least $2R - R = R$). Furthermore, conditioning on the additive noise being positive for $u$, there is a non-zero probability that $u$ has the smallest additive noise (in absolute value) among all pivots. $u$ is again ranked the highest in this case. Altogether, the probability that $u$ is ranked the highest in terms of the weak signal similarity is strictly larger than $1/2$. It follows that $\mathbb{E}[X] < \mathbb{E}[Y]$, as desired. $\quad\square$

### D.2 EXPLAINING WHY MERGING HELPS

We consider a particular worst case example for the `KwikCluster` algorithm which motivates why a post processing merging step helps. At a high level, it is possible to pick pivots which do not have a strong signal edge but nevertheless 'should' belong to the same cluster. Then when we a clustering algorithm is run, these two pivots can possibly lead to two disjoint clusters whereas that merging them lowers the correlation clustering objective and improves the overall clustering quality.

Concretely, consider the following example: we have a complete graph on $n$ vertices where every edge is a strong signal edge except a single edge $(u, v)$ which is not. In the classical `KwikCluster` algorithm, if $u$ is picked as a pivot then we will form two clusters, one consisting of all vertices besides $u$ and the the other cluster being the singleton $\{u\}$. The same is true if we pick $v$ to be the pivot. Thus the expected correlation clustering objective of the algorithm is

$$\frac{2}{n} \cdot (n - 1) + \left(1 - \frac{1}{n}\right) \cdot 1 \to 3.$$

On the other hand, clustering every vertex to be one cluster has correlation clustering objective value 1. Thus in the case where there are two clusters in the above example, a merging post-processing improves the overall cluster quality. This crisply captures our motivation.

While the above situation may not be representative, our merging post processing verifies that a possible merge is sound (after ranking possible cluster candidates to merge using the weak signal) by querying a (small) number of strong single values and merging only if the average strong single similarity is sufficiently high. Thus our post processing merge routine can only help the overall clustering.

### D.3 INHERENT TRADE-OFFS BETWEEN PRECISION AND RECALL

The goal of this section is to show that there is an inherent tradeoff between precision and recall of *any* clustering algorithm on graphs. We first restate the definitions of precision and recall as defined in our experimental section. Let $G$ be an unweighted (not necessarily complete) graph and let $\mathcal{C}$ be a clustering of its vertices. The edges of $G$ correspond to the edges in the strong signal graph (i.e., the edges are pairs of vertices the strong signal labels as 'similar.'). Correspondingly, the non-edges of $G$ represent the negative edges of the strong signal.

We first restate the definitions of precision and recall as defined in our experimental section. The recall of $\mathcal{C}$ is defined as the fraction of edges of $G$ which are together in some cluster given by $\mathcal{C}$. The precision of $\mathcal{C}$ is defined as the fraction whose numerator is the number of edges of $G$ which are together in some cluster and the denominator is the total number of pairs of vertices that are clustered together.

We state a natural (random) graph dataset such that with high probability, *any* clustering $\mathcal{C}$ has either recall or precision bounded away from 1 by a fixed constant. In particular, $G$ will be sampled from the standard $G(n, 1/2)$ Erdos-Renyi graph distribution. Note the order of the quantifiers: we first generate a random graph. There is an event $E$ which $G$ satisfies with high probability. Condition on this event, *any* clustering of the vertices of $G$ will either have its precision or recall bounded away from 1, including the OPT correlation clustering.

Note that we have not made an attempt to optimize the constants in the following lemma for clarity. It is likely that one can optimize our proof and obtain a smaller constant than 0.75.

**Lemma D.2.** *Let $G$ be sampled from the $G(n, 1/2)$ distribution. With probability at least $1 - 1/poly(n)$, **all** clusterings of $G$ have recall or precision at most* 0.75.

*Proof.* Let $C > 1$ be a fixed constant. We first consider the following event $E$ : for any subset $S$ of vertices of size at least $C \log n$, there are at most $1.01|S|^2/4$ and at least $0.99|S|^2/4$ edges of $G$ within $S$.

We now show that $E$ holds with probability at least $1 - 1/poly(n)$. For a fixed subset $S$ of $k$ vertices for $k \geq C \log n$, the expected number of edges within $S$ is $\binom{k}{2}/2$. Thus the probability that there are more than $1.01k^2/4$ and less than $0.99k^2/4$ edges within $S$ is at most $\exp(-ck^2)$ by a standard Chernoff bound for a fixed constant $c > 0$. There are $\binom{n}{k}$ such choices of $S$ and thus union bounding over all $S$ and all $k \geq C \log n$, we have that the probability there exists some set $S$ with $|S| \geq C \log n$ vertices violating the required number of edges is at most

$$\sum_{k \geq C \log n} \binom{n}{k} \exp(-ck^2) \leq \sum_{k \geq C \log n} 2 \left(\frac{ne}{k}\right)^k \exp(-\varepsilon^2 k^2/6)$$

$$= \sum_{k \geq C \log n} \exp(\log 2 + k \log(ne) - k \log k - ck^2)$$

$$\leq n \cdot \exp(-\Omega(k^2))$$

$$\leq \frac{1}{poly(n)}$$

for $k \geq C \log n$ for a sufficiently large constant $C$ and $n$ large enough. Thus $\mathbb{P}(E) \geq 1 - 1/poly(n)$ where we can make the polynomial arbitrarily large by increasing $C$. We also condition on the fact that $G$ itself has at least $0.999n^2/4$ edges with also happens with inverse polynomial failure probability.

Now consider an arbitrary clustering $\mathcal{C}$. If the recall of $\mathcal{C}$ is at most $0.75$ then we are done so suppose the recall is at least $0.75$. Given this, we claim that there exist a cluster within $\mathcal{C}$ of size at least $0.74n$.

To see this, let $C_1, \cdots, C_j$ be the clusters of $\mathcal{C}$. The clusters of size at most $C \log n$ have at most $n \cdot (C \log n)^2/2$ edges of $G$ inside them. All other clusters $C_i$ have at most $1.01|C_i|^2/4$ edges of $G$ inside them. Altogether, the number of edges of $G$ inside some cluster is at most

$$n \cdot (C \log n)^2/2 + \sum_{|C_i| \geq C \log n} \frac{1.01|C_i|^2}{4}$$

subject to the constraint that $|C_i| \leq 0.74n$ and $\sum_i |C_i| \leq n$. This is a convex function which is maximized at its boundary, meaning the number of edges of $G$ inside some cluster of $\mathcal{C}$ is at most

$$n \cdot (C \log n)^2/2 + \frac{1}{0.74} \cdot \frac{1.01 \cdot 0.74^2 n^2}{4} \ll 0.75 \cdot \frac{0.999n^2}{2}$$

which contradicts the fact that the recall of $\mathcal{C}$ is at least $0.75$. Thus there exists a cluster of $\mathcal{C}$ of size at least $0.74n$. Now given this, we show that the precision must be at most $0.75$.

Towards this end, let $C_i$ be the cluster of $\mathcal{C}$ of size at least $0.74n$. It has at most $1.01|C_i|^2/4$ edges of $G$ inside it and $\binom{|C_i|}{2}$ pairs of vertices. Let $A$ be the other edges of $G$ not inside $C_i$ and let $B$ be the total pairs of vertices where both of the vertices in the pair lie outside of $C_i$. Then the precision of $C_i$ is bounded by

$$\frac{1.01|C_i|^2/4 + A}{\binom{|C_i|}{2} + B}.$$

Since $|C_i| \geq 0.74n$, one can easily verify that

$$B \leq \frac{(0.26n)^2}{2} < \frac{1}{8} \cdot \binom{|C_i|}{2}$$

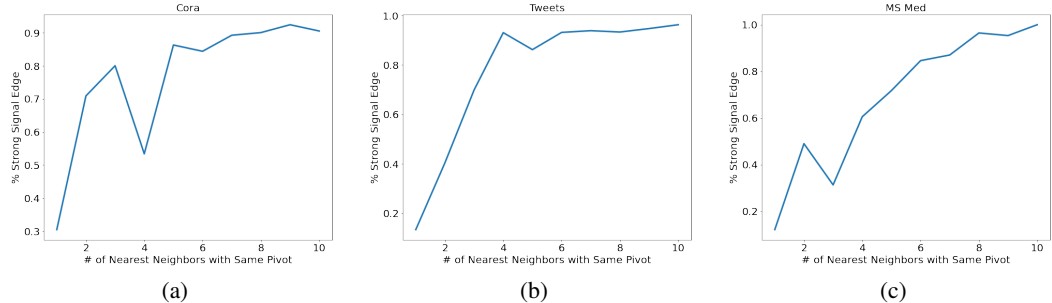

Figure 2: The propensity for a vertex $v$ to connect to a pivot $p$ given that $k$ of $v$'s neighbors have already connected to $p$.

and thus

$$
\begin{aligned}
\frac{1.01|C_i|^2/4 + A}{\binom{|C_i|}{2} + B} &\le \frac{1.01|C_i|^2/4 + B}{\binom{|C_i|}{2} + B} \\
&\le \frac{1.01|C_i|^2/4}{\binom{|C_i|}{2}} + \frac{B}{\binom{|C_i|}{2} + B} \\
&\le 0.51 + \frac{B}{9B} \\
&< 0.75,
\end{aligned}
$$

as desired. $\qquad\square$

## D.4 Motivating Using Weak Signal Neighborhood Statistics.

In Figure 2, we plot the the fraction of times a vertex $v$ connects to a pivot $p$ in the `KwikBucks` algorithm as a function of the number of nearest neighbors of $v$ (in terms of the weak signal similarity) which have already connected to the same pivot $p$.. We see that the probability increases as a function of the number of nearest neighbors, empirically justifying our algorithmic design optimization of 'Utilizing Weak Signal Neighborhood' in Section C. Note that this optimization has the affect of slightly boosting such pivots $p$ (if they exist) to a higher similarity (and thus a better ranking).

## E    Details for Dataset & Weak/Strong Signals

We provide a detailed description of the datasets used in the paper as well as the weak and strong signals used for each of the datasets. Table 2 provides a summary.

**Stackoverflow (SOF) and SearchSnippets:** Stackoverflow and SearchSnippets are commonly used for short-text clustering/classification. For stackoverflow, we used a subset collected by Xu et al. (2017) consisting of of 20,000 question titles associated with 20 different categories obtained from a dataset released as part of a Kaggle challenge. For SearchSnippets, we used the dataset from Phan et al. (2008) which consists of 12,340 snippets (extracted from web search snippets) associated with 8 groups. For these two datasets, we experimented with two different types of cheap signals: word2vec embeddings Mikolov et al. (2013) and tf-idf embeddings. In both cases, we trained/finetuned on the training set of the datasets. We used the Gensim package Rehurek & Sojka (2011) for word2vec and sklearn Pedregosa et al. (2011) for tf-idf. Word2vec provides a vector representation for each English word; to compute the embedding for a sentence/document, we average the embeddings of each of its words. For the strong signal, for each dataset we finetuned a T5-1.1 XXL model (11B parameters) Raffel et al. (2020) on the training data where given two examples, the model was finetuned to predict if they belong to the same cluster or not. In both cases, we sampled 10K positive pairs and 50K negative pairs and finetuned the model for 10 epochs on a 4x4 DragonFish TPU architecture.

**Twitter and AgNews:** Twitter and News data are commonly used for short-text clustering/classification. From Twitter, we use the dataset created by Yin & Wang (2016) consisting of 2,472 tweets with 89 categories. From News, we use the data from Rakib et al. (2020) which is a subset of the dataset from Zhang & LeCun (2015) containing 4 topics. For the cheap signal, we use pretrained BERT embeddings Devlin et al. (2018) where we feed each example into the BERT model, obtain contextual token embeddings, and then average them (ignoring the [CLS] and [SEP] tokens) to obtain the embedding for each example. We use the 12-layer uncased BERT-Base model for this experiment. For the strong signal, we first created a graph by connecting two nodes if they belong to the same category, then added noise to the graph by flipping the existence/non-existence of an edge for $5\%$ of node pairs selected uniformly at random (note that without adding noise, the problem becomes much easier as graph of the strong signal becomes composed of multiple connected components).

**Internal:** This is a vertical of a large, internal, proprietary text dataset. The weak signal is embedding similarity, and the strong is an indicator variable from a cross-attention model.

**Citeseer and Microsoft Medicine:** Citeseer Sen et al. (2008) and Microsoft Medicine Shchur & Günnemann (2019) are attributed graph datasets. Citeseer is a citation network in which nodes represent papers, edges represent citations, and features are bag-of-word abstracts. Microsoft Medicine is a subset of the Microsoft Academic graph where the nodes represent authors, edges represent co-authorship, and node features are a collection of paper keywords from author's papers. For both datasets, we used the cosine similarity between the node features as the weak signal and we assume the edges of the graph correspond to the strong signal.

**Cora and Amazon Electronics Photos:** Similar to Citeseer and Microsoft Medicine, Cora and Amazon Electronics Photos are also attributed graph datasets. They are typically used for node classification but here we adapt them to our problem. Cora Sen et al. (2008) is a citation network similar to the Citeseer dataset with the node labels corresponding to paper topics. Amazon Electronics Photos Shchur et al. (2018) is a subgraphs of the Amazon copurchase graph where the nodes represent goods, an edge between two nodes represents that they have been frequently purchased together, node features are bag-of-word reviews, and class labels are product categories. For these two datasets, we used the deep graph infomax (DGI) model Velickovic et al. (2019) to learn unsupervised node representations and used these representations as the cheap signal. We also used noisy labels as the strong signal similar to the Twitter dataset.

**Total cost analysis:** Our work is mostly based on the applications where the weak oracle values are computed via distances based on embeddings and the strong signal values are the output of a large cross-attention transformer model. In this case, there are three different factors that comprise the total cost of the clustering algorithm: 1- the cost of the queries to the strong signal, 2- the cost of computing embeddings from the cheap signal, and 3- the cost of geometric operations on the embeddings. So the total cost can be summarized as follows:

$$Total\ Cost = \eta_S \zeta_S + \eta_E \zeta_E + \eta_G \zeta_G$$

where $\eta_S$ represents the number of calls to the strong signal, $\zeta_S$ represents the cost of making a call to the strong signal, $\eta_E$ represents the number of calls needed to compute embeddings, $\zeta_E$ represents the cost of obtaining one embedding, $\eta_G$ represents the number of geometric operations (cosine similarity in our case) we perform on the embeddings, and $\zeta_G$ represents the cost of a single geometric operation.

The number of calls $\eta_E$ required to obtain embeddings is $n$ (i.e. the number of data points) which is smaller than $\eta_S$ (which, in our case, is typically a linear factor of $n$) and the cost $\zeta_E$ of obtaining one embedding is significantly smaller than the cost of obtaining one strong signal similarity $\zeta_S$. Therefore, $\eta_E \zeta_E$ can be subsumed in $\eta_S \zeta_S$.

When using 32 TPU v3 chips for the strong signal and a CPU for the geometric operations, each call to the strong signal was approximately $10^4$ times slower (i.e. $\zeta_S \approx 10^4 \zeta_G$). This gap becomes even more stark if we use fast geometric algorithms such as nearest neighbor search or use TPUs for geometric operations. It follows from the analysis of our algorithm that $\eta_G \in O(nk)$ where k is the parameter defined in Algorithm 6. This is comparable to $\eta_S$. Therefore, $\eta_G \zeta_G$ is negligible compared to $\eta_S \zeta_S$ in our experiments.

Following the above justifications, as well as for theoretical simplicity, in this paper we ignored the cost of querying the weak signal in our analysis (i.e. assume $\eta_E \zeta_E + \eta_G \zeta_G \approx 0$). However, if future

work considers costlier operations for the cheap signal, these extra terms should also be considered in determining the total clustering cost.

## F  ADDITIONAL EXPERIMENTAL RESULTS

### F.1  PRECISION AND RECALL

The precision and recall (with respect to a clustering $C$) definitions used in Section 4 are defined as follows:

$$\text{Precision}(C, O_S) = \frac{\sum_{e=(i,j)} C_{i,j} O_S(e)}{\sum_{e=(i,j)} C_{i,j}} \tag{2}$$

where $C_{i,j}$ is the indicator for if vertices $i, j$ are in the same cluster.

$$\text{Recall}(C, O_S) = \frac{\sum_{e=(i,j)} C_{i,j} O_S(e)}{\sum_{e=(i,j)} O_S(e)}. \tag{3}$$

As stated in (García-Soriano et al., 2020a), while our algorithm and baselines have been designed to minimize the total correlation clustering objective, it is important to consider precision and recall as they are problem independent measures of cluster quality. Furthermore in cases where the underlying strong signal graph is extremely sparse, the correlation cost objective might not be meaningful. For example in such a case, returning all vertices as singleton clusters already has low objective value (equation 1). We use the entire strong signal graph for the purposes of evaluating the experimental metrics, such as CC objective, precision, and recall.

Table 2: Properties of datasets used in our experiments. $n$ denotes the number of vertices and Non-zero entries denotes the number of non-zero entries in the adjacency matrix of the strong signal graph (i.e. twice the number of edges), both rounded to two significant digits.

| Name | Type | Weak Signal | Strong Signal | $n$ | Non-zero entries |
|------|------|-------------|---------------|-----|------------------|
| SOF | Text | W2V / tf-idf | Cross-attention model | $4.9 \cdot 10^3$ | $2.3 \cdot 10^6$ |
| Search | Text | W2V / tf-idf | Cross-attention model | $3.3 \cdot 10^3$ | $2.0 \cdot 10^6$ |
| Twitter | Text | BERT Embeddings | Noisy label indicator | $2.4 \cdot 10^3$ | $4.7 \cdot 10^5$ |
| AgNews | Text | BERT Embeddings | Noisy label indicator | $8.0 \cdot 10^3$ | $1.8 \cdot 10^7$ |
| Internal | Text | Embeddings | Cross-attention model | $1.0 \cdot 10^5$ | $9.5 \cdot 10^7$ |
| Cora | Attributed Graph | DGI Embeddings | Noisy label indicator | $2.7 \cdot 10^3$ | $1.5 \cdot 10^6$ |
| Photos | Attributed Graph | DGI Embeddings | Noisy label indicator | $7.7 \cdot 10^3$ | $1.2 \cdot 10^7$ |
| Citeseer | Attributed Graph | Node Features | Adjacency matrix | $3.3 \cdot 10^3$ | $10^4$ |
| Med. | Attributed Graph | Node Features | Adjacency matrix | $6.3 \cdot 10^4$ | $1.6 \cdot 10^6$ |

### F.2  PARAMETER SELECTION DETAILS

We first describe how to select the value $t$ in Algorithm 3 and $k$ in Algorithm 6, which selects the top $k$ vertices in weak-signal similarity for the strong signal to query.

The intuition in picking $t$ is that it must be sufficiently large so that only few vertices do not have a pivot in their neighborhood (and thus contribute to the additive error of Theorem 2.1). This parameter naturally depends on the density of the underlying strong signal graph: for sparser graphs, one must pick a larger value of $t$ since each vertex on average has a small degree and is thus less likely to have a pivot chosen in its neighborhood than a vertex with a larger degree. We use the above intuition to design the following data-dependent method to select $t$: we first sample a sublinear number of random strong signal edges ($\sqrt{n}$ strong signal edges to be exact). This returns an estimate of the density of the graph up to small additive error (for example via standard Chernoff bounds). We then set $t$ to be 10 times the inverse of the density. If the density is extremely sparse, i.e. less than $1/1000$ fraction of possible edges exist, we simply set $t$ to be equal to $n/2$.

The second parameter we set is $k$ in WeakFilterByRanking. We can pick a value of $k \ll n$ because intuitively, a meaningful weak signal assigns a high similarity score to relevant pivots

relative to all other pivots and thus such pivots have higher ranking. To understand the trade offs in selecting $k$, consider the most prominent place where it is used in our algorithm: when a vertex $v$ attempts to find a strong signal edge to one of the pivots by iterating through them in the weak signal ordering. The trade offs are the following: a smaller value of $k$ leads to better query efficiency as $v$ is guaranteed to only make $k$ strong signal queries in this step. However the clustering quality can suffer because the first $k$ pivots, for a small $k$, in the weak signal order might not have a strong signal edge to $v$. Conversely a larger value of $k$ leads to increased exploration from $v$ as it attempts to connect to a pivot. However in the case that $v$ is truly a singleton cluster, i.e. it has no strong signal edges to any pivot, we potentially waste many strong signal queries. To balance these trade offs, we pick an 'intermediate' value of $k = 100$ *for all our experiments*. Ablation studies for both parameters are given in Section F.

We also always set $k = 10$ when we use the "Utilizing Weak Signal Neighborhood" optimization of Section 3. We also always fix $\lambda = 1/10$ which appropriately normalizes the second term to be between 0 and 1 (note the weak signal similarity $w_p$ is between $-1$ and 1). The parameter 10 here is fairly robust and can likely be replaced by any (small) reasonable value and we also perform ablation studies on this optimization.

For spectral clustering, we always use $k = 25$ for the number of clusters. Higher values were computationally prohibitive to use.

## F.3 RESULTS

We present additional experimental results in Figure 3 and 4 which show similar qualitative results as Figure 1: our algorithm `KwikBucks` has superior query complexity over the baselines as it achieves a higher $F_1$ value (and lower CC objective values) while utilizing fewer strong signal queries than baselines.

Table 3: CC objective values are shown for a fixed budget of $3n$. See Table 1 for the corresponding $F_1$ values. We normalize the smallest CC value to 1.0 so smaller quantities are desirable. See Figures 3 and 4 for results as a function of query budget. For the sparser graph datasets of Citeseer, Med., and Internal we use the budget of $50n$. Due to their sparsity, the CC objective value is less meaningful than $F_1$ values for these two datasets.

|  | SOF | Search | Tweet | AgNews | Cora | Photos | Citeseer | Medicine | Internal |
|---|---|---|---|---|---|---|---|---|---|
| B1 | 1.9 | 2.5 | 1.2 | 2.0 | 2.0 | 2.5 | 1.3 | 1.1 | 1.01 |
| B2 | 1.8 | 2.0 | 1.2 | 2.0 | 2.0 | 2.4 | 1.3 | 1.1 | 1.04 |
| B3 | 6.4 | 4.0 | 6.3 | 2.5 | 2.5 | 2.2 | 745.1 | 2550.8 | - |
| B4 | 2.0 | 6.0 | 1.1 | 4.1 | 3.0 | 3.2 | 1.0 | 1.0 | 1.01 |
| B5 | 2.0 | 6.0 | 1.1 | 4.1 | 3.0 | 3.2 | 1.3 | 1.3 | 1.01 |
| **KwikBucks** | 1.0 | 1.0 | 1.0 | 1.0 | 1.0 | 1.0 | 1.1 | 1.0 | 1.0 |

## F.4 ADDITIONAL ABLATION RESULTS

In our ablation experiments, we fix all parameter settings except the component we are altering. We perform ablation studies on 4 representative datasets: Cora, Citeseer, Stackoverflow (SOF), and Search. Our first observation is that the merge post processing procedure can help return a higher quality clustering, for example for the Cora, SOF, and Citeseer datasets; see Figures 5 and 6 for details and Section D.2 for theoretical intuition of why post processing merging helps.

Next we consider removing the `SortNonPivots` step and replacing it with an using an arbitrary ordering of non pivot vertices. We see that the positive benefits of removing this component are more subdued compared to the merge post processing. However, this change never hurts the quality of the clustering. Overall, we view the different data-driven components introduced in Section 3 and C as having complementary benefits as each optimize a different part of the algorithm.

We observe that one must choose a sufficiently large value of $t$ in `GetPivots` which is the initial number of random vertices sampled which are later processed to be pivots. As argued in Section F.2, it is important to select a sufficiently large value of $t$ to limit the number of vertices which do not have a pivot in their strong signal neighborhood (as captured by the additive error term in Theorem

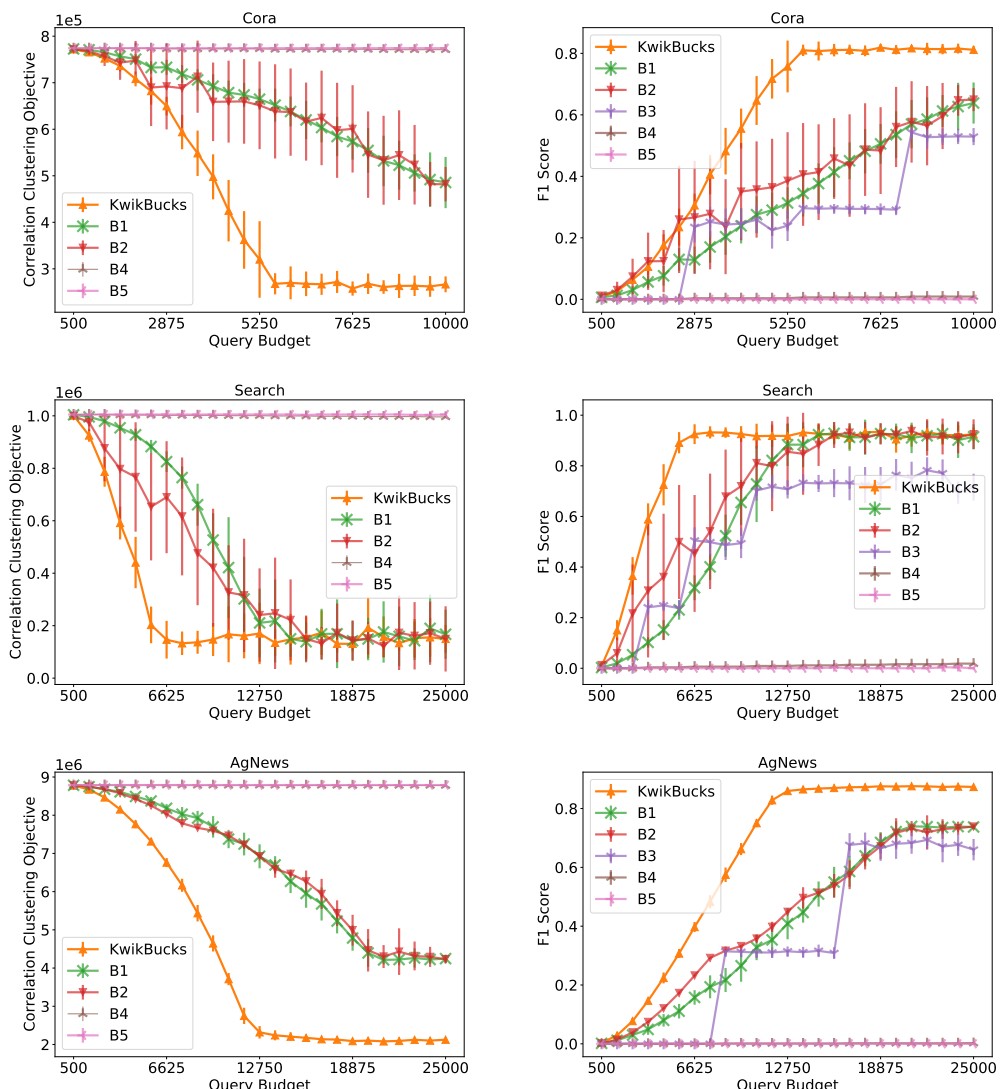

Figure 3: Empirical results for the Cora, Stackoverflow, Search, and News datasets.

2.1). For our ablation studies, we consider two other settings of $t$, one which is a factor of 10 smaller than the choice used in our main experimental results and one which is a factor of 10 larger. They represent the 'Small' and 'Large' pivot choices respectively. We see in Figures 5 and 6 for the Cora and Citeseer datasets, that a smaller choice of $t$ can lead to a decrease in the performance of our algorithm. Nevertheless, our data driven density based approach outlined in F.2 hits the 'sweet spot' and performs comparable to the best choice of pivots in all cases as shown in Figures 5 and 6.

We also perform ablation experiments on the choice of $k$ in `WeakSignalFilterPractice` by considering $k = 10$ and $k = 1000$ (a 'small' and 'large' choice respectively as before). Our ablation experiments also show that a large choice of $k$ in `WeakSignalFilterPractice` can lead to many queries wasted as argued in Section F.2. Indeed, we see in the above figures that for the Citeseer dataset, a large value of $k$ leads to worse performance initially as we waste many strong signal queries on vertices which have no strong signal edge to any of the pivots. This is due to the sparse nature of the Citeseer dataset. However as the query budget is increased, the quality of the clustering improves. The choice of $k$ seems to have negligible impact on the other datasets we tested on and our choice of $k = 100$ (which we fixed in the main experimental results) was always competitive.

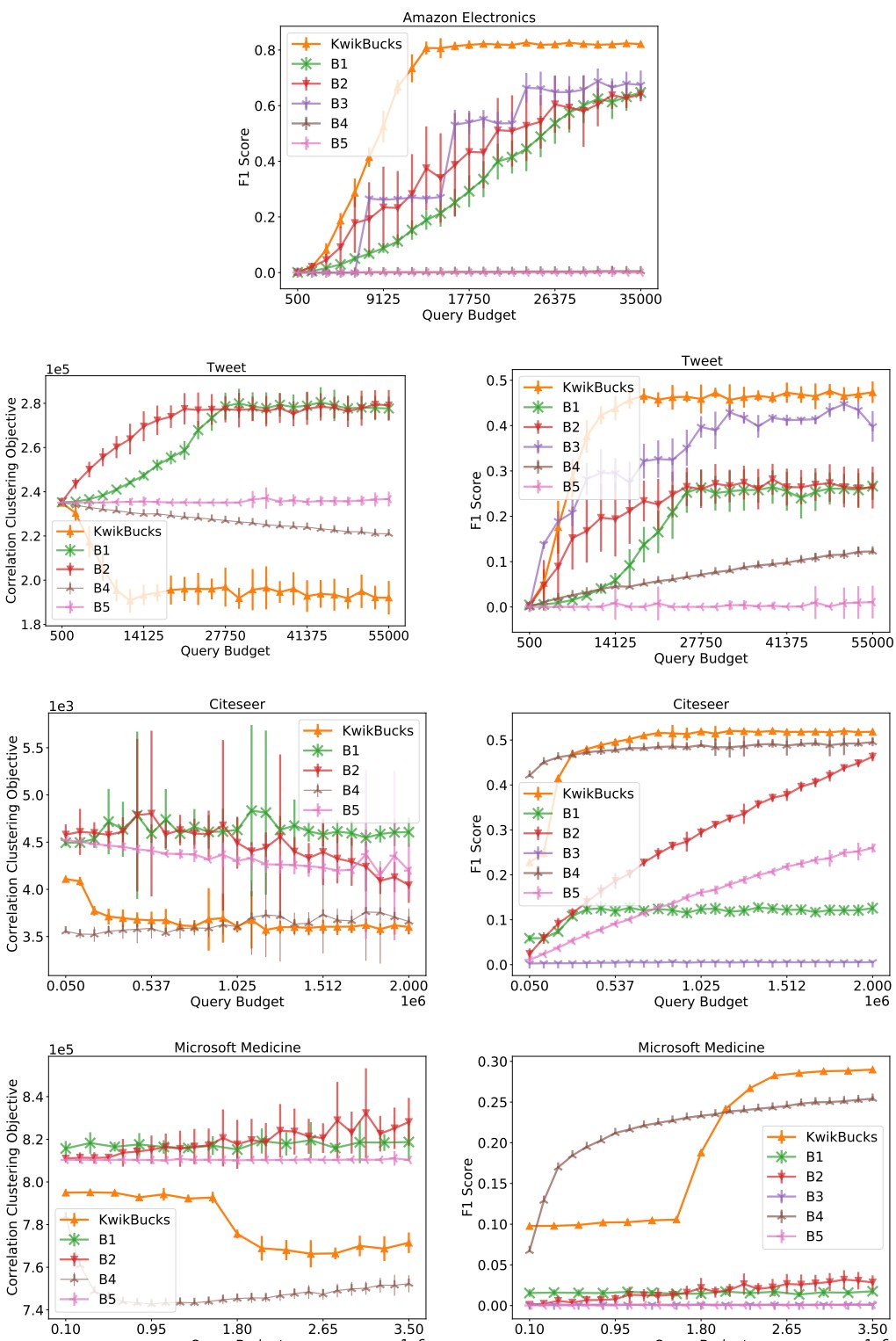

Figure 4: Empirical results on the datasets omitted from Figure 3. The results are qualitatively similar to that of Figure 3.

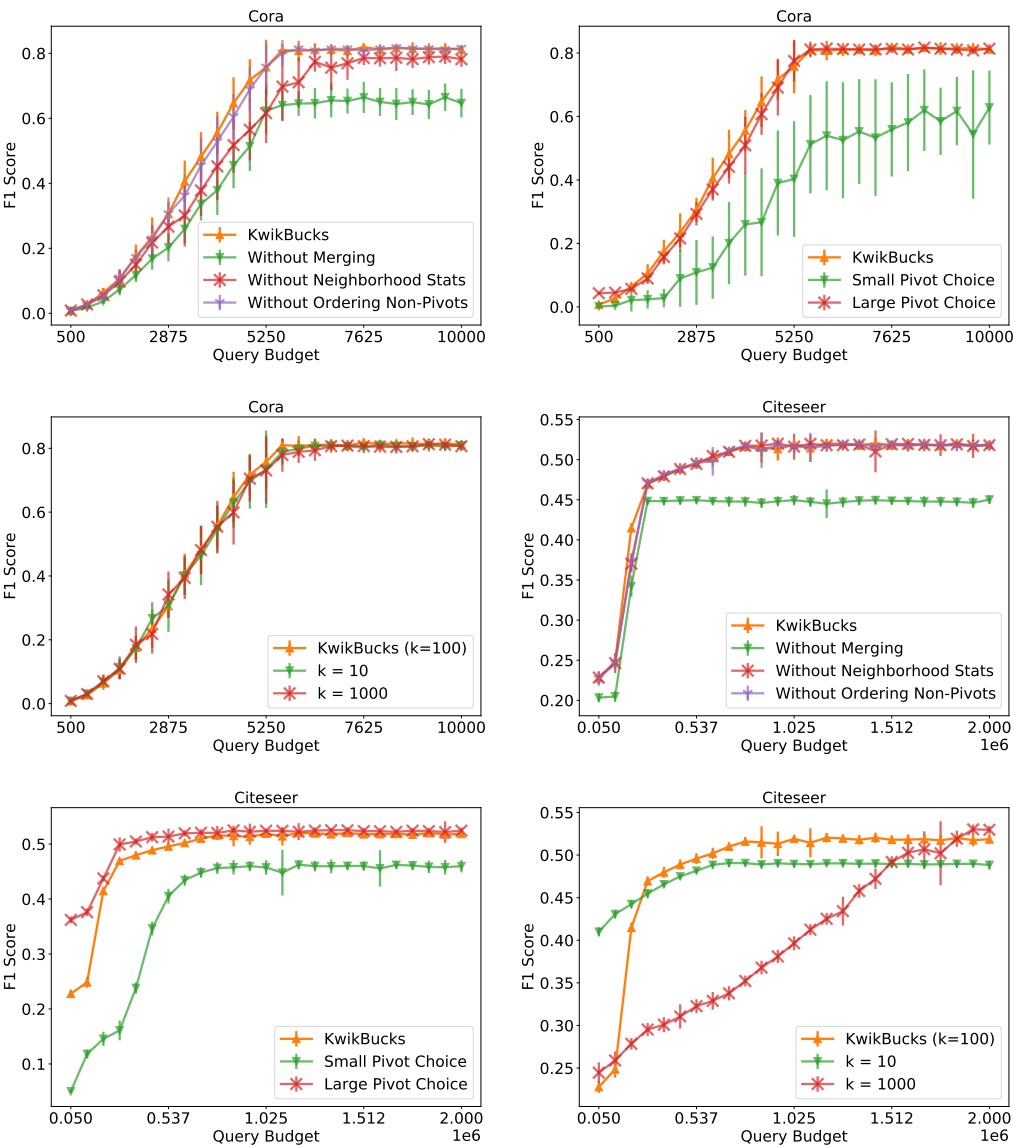

Figure 5: Figures for ablation studies for Cora and Citeseer datasets.

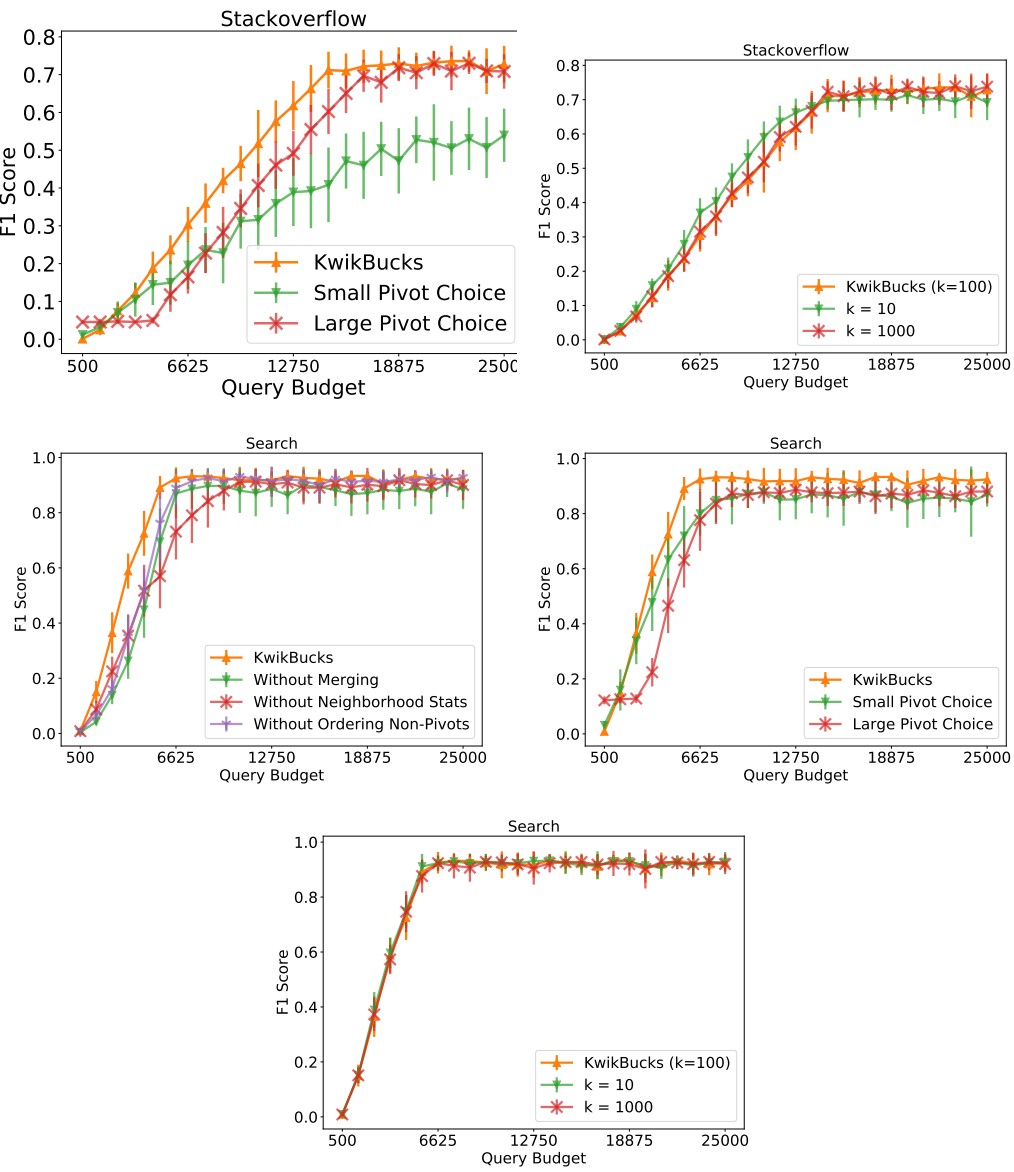

Figure 6: Figures for ablation studies for SOF and Search datasets.

## F.5 MEASURING THE QUALITY OF WEAK SIGNALS

We design a simple and informative experiment to measure the quality of weak signals. For the Stackoverflow (SO) dataset, we run `KwikBucks` where we replace the weak signal with a linear *interpolation* of the strong signal and a random matrix will all entries i.i.d. from the uniform distribution in $[-1, 1]$. The purpose of this experiment is to show a higher quality weak signal gives better clustering results than using a lower quality weak signal. Indeed, Figure 8 shows that `KwikBucks` performs the best if we replace the weak signal completely with the strong signal, as naturally expected. As we vary the amount of randomness in the weak signal, the performance degrades and the case where the weak signal is a fully random matrix performs the worst as a function of query budget. It is also interesting to consider the cases where the weak signal are given by the (stronger) W2V model versus the comparatively weaker tf-idf model: the performance of using the W2V embeddings for the weak signal lies between the 'half-random' and '2/3 random' case whereas the tf-idf plot lies between the '2/3 random' and 'fully random' cases. The random interpolated weak

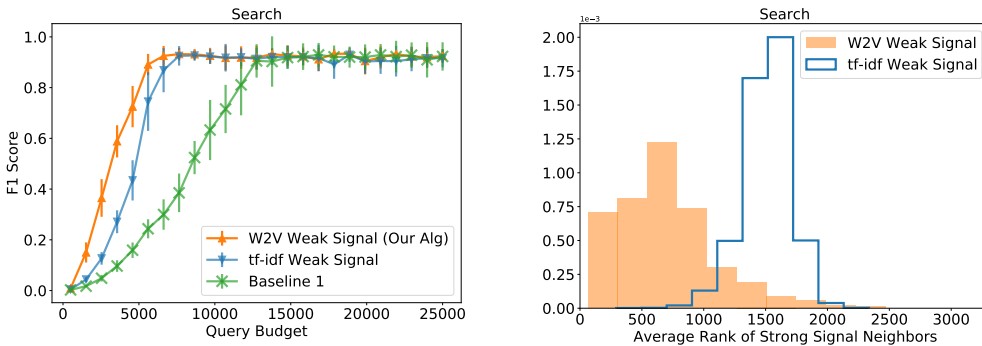

Figure 7: The figure shows a qualitatively similar result as the SOF results shown in Figure 1.

signal cases, while artificial, help us qualitatively access the usefulness of a particular real world weak signal instance.

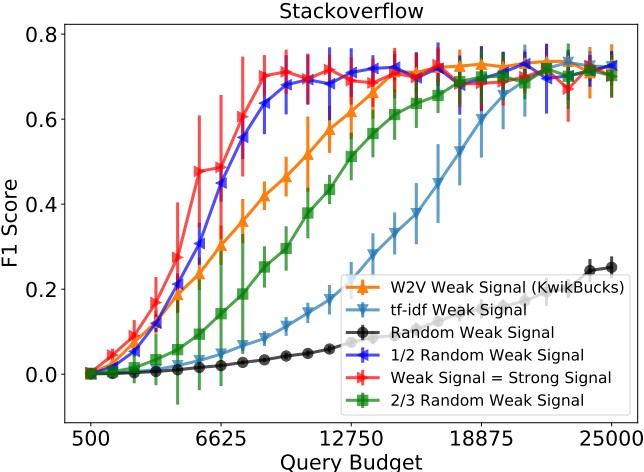

Figure 8: Interpolating the weak signal between uniformly random values and the strong signal.

## F.6 Average Rankings of Strong Signal Neighbors

In this Section we present additional experiments in the similar spirit as the right figure of the second row of Figure 1 for the Tweet, Med., and Cora datasets. For every vertex $v$ in these datasets, we rank all the other vertices in decreasing weak signal similarities to $v$. The average rank of the true strong signal neighbors of $v$ is computed and plotted as a histogram (normalized to be a distribution). Intuitively, a good weak signal should have the property that *true* strong signal neighbors have much higher weak signal similarity scores (and thus better rankings) than the an arbitrary vertex. Indeed, we see that to be the case of the datasets in Figure 9 where the distributions are much more left shifted and has a much smaller mean compared to the case if the weak signal was fully random. This validates the connection between our empirical weak signal Definition 1.3 and the theoretical assumption we made for the weak oracle in Assumption 2.1. Indeed, Figure 9 gives empirical validation to the claim that returning a top $k$ most similar vertices to a vertex $v$ in terms of weak signal similarity captures many actual *true* strong signal neighbors.

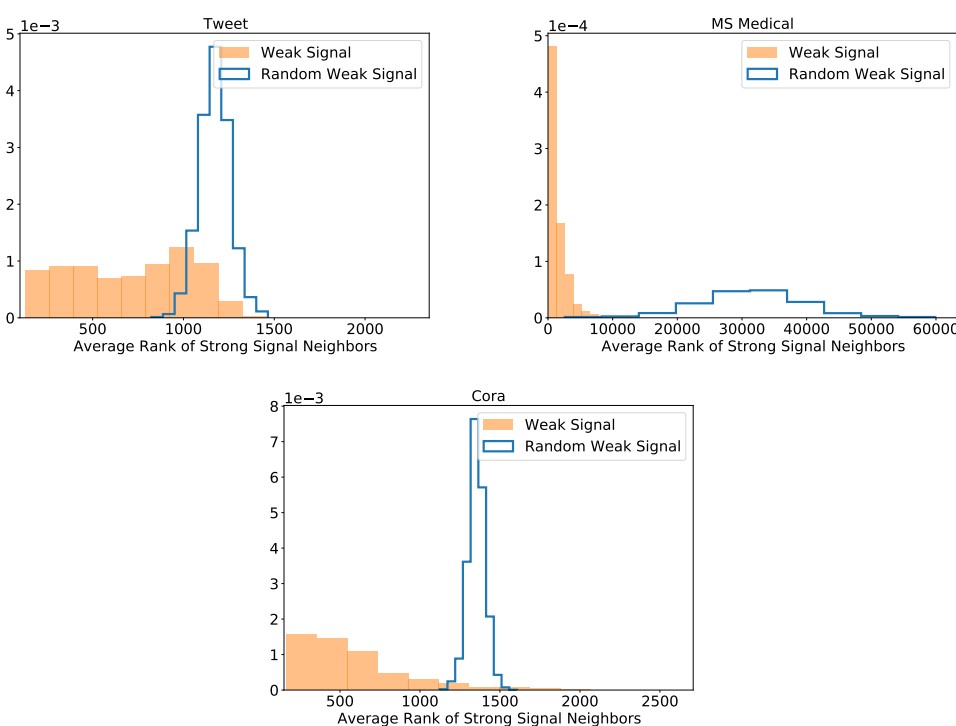

Figure 9: The average weak signal rank of actual strong signal neighbors is shown in orange. The blue curve shows the average rank if the weak signal was fully random.

