# OpenReview forum: "KwikBucks: Correlation Clustering with Cheap-Weak and Expensive-Strong Signals"
_ICLR.cc/2023/Conference — ICLR 2023 poster_

### Official Review · Reviewer_6pez · 2022-10-23

**Confidence:** 4
**Clarity, Quality, Novelty And Reproducibility:** Please refer to Summary Of The Review.
**Correctness:** 3
**Technical Novelty And Significance:** 3
**Empirical Novelty And Significance:** 3
**Recommendation:** 6

**Strength And Weaknesses:**

Please refer to Summary Of The Review.

**Summary Of The Paper:**

Please refer to Summary Of The Review.

**Summary Of The Review:**

In this study, the author(s) propose(s) a novel correlation clustering algorithm named KwikBucks. It’s an extension of a well-known correlation clustering algorithm KwikCluster with the consideration of the limited query budget for reliable pairwise similarity. So the designed KwikBucks algorithm less the queries for expensive-strong signal by adding cheap-weak signal acquisition steps. The performance of the KwikBucks algorithm was tested by 9 datasets with different properties (having density changes of strong signal and weak signal). The experimental results shows KwikBucks can get a lower objective function value and higher F1 scores. Besides, it obtain 3-pproximation more efficiently.

The topic of this paper is meaningful as the problem of high cost of obtaining correlations among samples does exist because of the high computation of some machine learning models. Meanwhile, the theories about this work is sufficient. As a reader, I summarize some detailed issues as follows.

1) The introduction section overemphasizes the high cost of some machine learning models in obtaining the similarity between points, and neglects the in-depth analysis of the existing works and problems related to this paper. So the motivation and innovation of this paper are not presented well.
2) On page 1, there is too much analysis of experimental results so that parts of section 4 are repeated. Please just summarize in short words.
3) I believe that when we talk about correlation clustering, it is necessary to mention the research of Bansal (for ex. Bansal N, Blum A, Chawla S. Correlation clustering. Machine learning, 2004, 56(1):89–113).
4) On page 3, the differences with the existing work are not clearly explained. For example, how does the algorithm proposed by Guha et al. differ from the work in terms of obtaining information from multiple sources? What are the advantages and disadvantages of the existing work?
5) Section 1.2 has an extra ‘we use’ in the first paragraph.
6) What is the basis for Assumption 2.1? Also, what is the significance of the \gamma parameter? Why is it not specifically analyzed through experiments to explain its influence on the proposed algorithm in section 4, or explain this parameter theoretically?
7) The statement on page 8 that for many datasets such as Cora and Search, B1 and B2 are the best among our five baselines is not rigorous because only B2 is the best among five baselines on Cora and Search datasets.

In general, this paper needs more carefully revisions before fully accepting it.

---

> ### Author Response · Authors · 2022-11-11
> **Thank you to reviewer 6pez. Response #1.**
>
> > The introduction section overemphasizes the high cost of some machine learning models in obtaining the similarity between points, and neglects the in-depth analysis of the existing works and problems related to this paper. So the motivation and innovation of this paper are not presented well.
>
> Thank you for this feedback. The high cost of machine learning models is the main inspiration for the paper and is an important and topical problem facing the ML community at large due to the resources required to train and inference ever growing models, especially in the NLP domain. Therefore, we believe the problem is well motivated and timely. In terms of existing works, we have added a more comprehensive discussion of the Guha et al. paper; see the response below. Is there any specific point that you would like us to discuss with regards to motivation or innovation?
>
> > I believe that when we talk about correlation clustering, it is necessary to mention the research of Bansal (for ex. Bansal N, Blum A, Chawla S. Correlation clustering. Machine learning, 2004, 56(1):89–113).
>
> Due to space limitations, we kept our focus on the application of correlation clustering, however we agree that we should give more context about the research in this area and so we rephrased the section to better cover prior research. Thanks for the suggestion!
>
> > On page 3, the differences with the existing work are not clearly explained. For example, how does the algorithm proposed by Guha et al. differ from the work in terms of obtaining information from multiple sources? What are the advantages and disadvantages of the existing work?
>
> While at a high level both Guha et al. and our work aggregate information across various signals, the two works differ in terms of the generality of oracles considered, the formal guarantees given, and the problems studied. The oracles used in Guha et al. are highly specialized to the datasets at hand; for example, the cheap oracle used in Guha et al. is an inverted index model which heavily relies on the specifics of the datasets used. In contrast, we take a broader view of weak and strong oracles and present theoretically founded algorithms which only assume query access to the weak and strong model and not any particular model idiosyncrasies. Therefore, our algorithm has provable guarantees on both the approximation quality and the query complexity, making it broadly applicable across different oracles. In terms of problems, we study correlation clustering while the focus of Guha et al. is not on a clustering problem. Rather, they use hierarchical clustering as an intermediate problem to perform user modeling and do not consider any specific clustering objective functions. The strong signal queries made by our algorithm are guided through formal reasoning and they exploit the structure of the clustering problem we are studying. In Guha et al. the weak signal is used at a more intuitive level and serves the informal role of filtering possible strong signal queries with no formal reasoning. Given these points, we conclude that while the two works have similar motivations, the work of Guha et al. is not directly applicable to our work. We have added this discussion to the additional related works section in Appendix A (highlighted in blue).
>
> > Section 1.2 has an extra ‘we use’ in the first paragraph.
>
> Thank you, the typo has been corrected.

---

> > ### Author Response · Authors · 2022-11-11
> > **Response #2.**
> >
> > > What is the basis for Assumption 2.1? Also, what is the significance of the \gamma parameter? Why is it not specifically analyzed through experiments to explain its influence on the proposed algorithm in section 4, or explain this parameter theoretically?
> >
> > The paper analyzes the significance of gamma both theoretically and empirically. Theoretically, as stated in the text around Assumption 2.1, $\gamma$ represents a theoretical modeling of the error of the weak signal. It nicely interpolates between the case where the weak signal is meaningless (i.e. $\gamma = n$) and when it is exact, i.e. $\gamma = 1$. The derived bound on the theoretical number of strong signal queries in Theorem 2.1 directly reflects this tradeoff. Furthermore, we prove in Lemma B.7 in the appendix that any algorithm *must* have at least the same dependence on $\gamma$ as in Theorem $2.1$.
> >
> > Empirically, we observed that the simple procedure of returning the most similar vertices for an input node captures many of the true strong signal neighbors of v and mimics the clean abstraction of Assumption 2.1. This claim is experimentally verified in Appendix F.6 and Figure 7 which shows that on average, the weak signal assigns true strong signal neighbors higher values. Furthermore, Figures 1(c) and Figure 8 display the effect of using progressively noisier weak signals. This can be interpreted as measuring the empirical impact of increasing the value of $\gamma$. As our theory predicts, a noisier weak signal, i.e. a higher value of $\gamma$, requires more strong signal queries to obtain the same cluster quality as a less noisy weak signal. Please see Appendix F.5 for further discussions on the effects of varying the quality of the weak signal which intuitively translates to varying the $\gamma$ parameter.
> >
> > > The statement on page 8 that for many datasets such as Cora and Search, B1 and B2 are the best among our five baselines is not rigorous because only B2 is the best among five baselines on Cora and Search datasets.
> >
> > Thank you for pointing this out. We have updated the text.
> >
> > We hope we have addressed your concerns and we would be happy to engage in further discussions. If you find our response satisfactory, we hope you will consider raising your score.

---

> ### Author Response · Authors · 2022-11-16
> **Follow up to response.**
>
> Dear Reviewer 6pez,
>
> Did we address all your concerns satisfactorily? If your concerns have not been resolved, could you please let us know which concerns were not sufficiently addressed so that we have a chance to respond before the November 18 deadline?
>
> Many thanks,
> The authors

---

### Official Review · Reviewer_Mf8c · 2022-10-27

**Confidence:** 3
**Correctness:** 4
**Technical Novelty And Significance:** 3
**Empirical Novelty And Significance:** 4
**Recommendation:** 6

**Clarity, Quality, Novelty And Reproducibility:**

Paper is written clearly and the setting considered is original and interesting in theory and practice.

**Strength And Weaknesses:**

I like the paper, and think the contribution is interesting. The experimental results are very good.

The paper focuses on the number of queries made to the strong oracle. However, the theoretical number of queries made to the weak oracle is reported nowhere. I think this is important, as the runtime of the algorithm might be end up being governed by the number of queries to the weak model if it is indeed queried many times.

The paper also claims (or cites) that correlation clustering is "perhaps the most natural formulation of clustering". I feel this is a subjective statement, and is better off not in the paper.

This leads to the question of designing an algorithm to optimize $Q_1 + \alpha Q_2$ where $Q_1$ is the number of queries to the strong model, and $Q_2$ is the number of queries to the weak model. $\alpha = 0$ is the setting studied in this paper, $\alpha = 1$ corresponds to the standard correlation clustering setting (since given a choice, one would always query the stronger model over the weaker one).



**Summary Of The Paper:**

The paper studies a version of correlation clustering where the learner has access to a weak predictor of node similarity in addition to an accurate stronger predictor. The weak model is assumed to have negligible query cost, and can be used in addition to the stronger model. For a free parameter $\epsilon$, he paper proposes a new algorithm which uses both the weak and strong oracle to retrieve a clustering which has objective value at most $3 \text{OPT} + \epsilon n^2$, and makes $O (d \gamma / \epsilon)$ queries to the strong oracle - here $\text{OPT}$ is the optimal objective value, $n$ is the number of nodes in the graph, $d$ is the average degree of the graph, and $\gamma$ captures the approximation power of the weak oracle compared to the strong oracle. The algorithm is implemented in practice and shown to have significant improvements in practical performance compared to standard baselines - a 64% relative improvement in clustering quality (in terms of F1 score) averaged over 9 datasets, and over > 3.5x reduction in number of queries made to the strong predictor compared to the best baseline.

**Summary Of The Review:**

I think the paper considers an interesting problem, and I think it is a fit for ICLR.

---

> ### Author Response · Authors · 2022-11-11
> **Thank you to reviewer Mf8c.**
>
>  > However, the theoretical number of queries made to the weak oracle is reported nowhere. I think this is important, as the runtime of the algorithm might be end up being governed by the number of queries to the weak model if it is indeed queried many times.
>
> Our work is mostly based on the applications where the weak oracle values are computed via distances based on embeddings and the strong signal values are the output of a large cross-attention transformer model. In this case, there are three different factors that comprise the total cost of the clustering algorithm: 1- the cost of the queries to the strong signal, 2- the cost of computing embeddings from the cheap signal, and 3- the cost of geometric operations on the embeddings. So the total cost can be summarized as follows:
>
> $$\text{Total Cost} = \eta_{S}\zeta_{S} + \eta_{E}\zeta_{E} + \eta_{G}\zeta_{G}$$
>
> where $\eta_{S}$ represents the number of calls to the strong signal, $\zeta_{S}$ represents the cost of making a call to the strong signal, $\eta_{E}$ represents the number of calls needed to compute embeddings, $\zeta_{E}$ represents the cost of obtaining one embedding, $\eta_{G}$ represents the number of geometric operations (cosine similarity in our case) we perform on the embeddings, and $\zeta_{G}$ represents the cost of a single geometric operation.}
>
> The number of calls $\eta_{E}$ required to obtain embeddings is $n$ (i.e. the number of data points) which is smaller than $\eta_{S}$ (which, in our case, is typically a linear factor of $n$) and the cost $\zeta_{E}$ of obtaining one embedding is significantly smaller than the cost of obtaining one strong signal similarity $\zeta_{S}$. Therefore, $\eta_{E}\zeta_{E}$ can be subsumed in $\eta_{S}\zeta_{S}$.
>
> When using 32 TPU v3 chips for the strong signal and a CPU for the geometric operations, each call to the strong signal was approximately $10^4$ times slower (i.e. $\zeta_{S}\approx 10^4 \zeta_{G}$). This gap becomes even more stark if we use fast geometric algorithms such as nearest neighbor search or use TPUs for geometric operations. It follows from the analysis of our algorithm that $\eta_{G} \in O(nk)$ where k is the parameter defined in Algorithm 6. This is comparable to $\eta_{S}$. Therefore, $\eta_{G}\zeta_{G}$ is negligible compared to $\eta_{S}\zeta_{S}$ in our experiments.
>
> Following the above justifications, as well as for theoretical simplicity, in this paper we ignored the cost of querying the weak signal in our analysis (i.e. assume $\eta_{E}\zeta_{E} + \eta_{G}\zeta_{G} \approx 0$). However, if future work considers costlier operations for the cheap signal, these extra terms should also be considered in determining the total clustering cost. We have added this discussion in Section E.
>
> > The paper also claims (or cites) that correlation clustering is "perhaps the most natural formulation of clustering". I feel this is a subjective statement, and is better off not in the paper.
>
> The statement has been updated.
>
> We hope we have addressed your concerns and we would be happy to engage in further discussions. If you find our response satisfactory, we hope you will consider raising your score.

---

> ### Author Response · Authors · 2022-11-16
> **Follow up to response.**
>
> Dear Reviewer Mf8c,
>
> Did we address all your concerns satisfactorily? If your concerns have not been resolved, could you please let us know which concerns were not sufficiently addressed so that we have a chance to respond before the November 18 deadline?
>
> Many thanks,
> The authors

---

### Official Review · Reviewer_neu8 · 2022-11-03

**Confidence:** 3
**Correctness:** 4
**Technical Novelty And Significance:** 3
**Empirical Novelty And Significance:** 3
**Recommendation:** 8

**Clarity, Quality, Novelty And Reproducibility:**

The model proposed is of incredible interest, especially due to the increasing use of large-scale ML models to predict the similarity between entities. Correlation clustering, being a very popular clustering paradigm with several applications (especially as a pre-processing step), is a suitable problem to be studied under this framework.

**Strength And Weaknesses:**

The proposed model (with strong-expensive and weak-cheap oracles) is powerful and pragmatic, in the sense that it allows one to navigate the trade-offs imposed due to the computationally-expensive process of obtaining the actual similarities between entities in real-world datasets. Further, this model helps one to leverage the existence of weaker-yet-useful models for predicting similarities to obtain more scalable algorithms. The theoretical results obtained complement the known results for the original version of the problem, except for the additive error term. The algorithm is simple and efficient (and as demonstrated in previous work has scope for parallelism as well.) The set of experiments performed is well-motivated and supports all the claims of the paper well.

**Summary Of The Paper:**

In this work, the authors introduce and study an interesting and pragmatic variant of the popular Correlation Clustering (Minimizing Disagreements) problem. In this problem, we are given a limited budget on the number of queries allowed to make to an expensive oracle that given an edge returns the true labeling of the edge (returned by possibly a large ML model). Moreover, we are also given unlimited access to a cheap but less accurate second oracle (for example embedding-based models, etc.). The authors propose an extension of the well-known KWIKCLUSTER algorithm, that uses the cheaper oracle wisely to reduce the number of queries to the expensive oracle. They prove a matching (to the original version of the problem) 3-approximation for a stronger assumption on the cheap oracle, except for an additive error. They show that the additive error term is unavoidable by demonstrating a tight lower-bound instance. They extend their algorithm to work for the more practical assumptions on the cheaper oracle, but with no theoretical guarantees. An extensive empirical analysis is performed to motivate the problem formulation, demonstrate the superior performance of their algorithm over baselines, and ablation studies on all tunable parameters of their algorithm.

**Summary Of The Review:**

Overall, this is a very well-written paper. The problem considered is of significant importance, the proposed algorithm is simple and efficient, and the empirical evaluation is extensive and supports the claims well. Therefore, I think this paper is a good fit for ICLR. Following are a few minor line-by-line comments:

Minor Comments:
- Page 2: Sec 1.1, para 2, line 1 - cluster -> clustering
- Page 3: para 1, line 4 - queries in for -> queries for
- Page 3: Sec 1.2, para 1, line 5 - “we use” is written twice
- Page 5: Thm 2.1 Proof Sketch, Line 1: inocrrectly -> incorrectly
- Page 6: Last para of Section 3, Line 10 - two cluster -> two clusters

---

> ### Author Response · Authors · 2022-11-11
> **Thank you to reviewer neu8.**
>
> > a few minor line-by-line comments
>
> We thank the reviewer for pointing out the typos. They have been corrected in the updated version of the paper.

---

### Author Response · Authors · 2022-11-11
**Thank you to all reviewers.**

We thank the reviewers for their valuable feedback. Answers are given in a response to each review. We have uploaded a revised version of the paper, with newly added segments marked in blue for the convenience of the reviewers.

---

### Author Response · Authors · 2022-11-18
**Update check**

Dear reviewers,

As the discussion period draws to a close, we wanted to check whether the reviewers or the area chair had any additional questions, comments, or feedback that we could address. We have also updated the paper incorporating reviewer feedback with text changes highlighted in blue for clarity.

Thank you again for your time and consideration!

---

### Author Response · Authors · 2022-12-08
**Discussion of Paper**

Dear Reviewers and Chairs,

We sincerely thank you for your efforts in reviewing the paper. Since the discussion period is drawing to a close, we would greatly appreciate it if you could read our responses and provide us with feedback. This will give us an opportunity to address any further questions you may have.

Many thanks,
The authors

---

### Decision · Program_Chairs · 2023-01-20

**Decision:**

Accept: poster

**Justification For Why Not Higher Score:**

The paper does not rise to the level of the oral/spotlight papers that I have seen at ICLR and other top-tier ML/AI conferences.

**Justification For Why Not Lower Score:**

The novelty and theory + experimental results are quite substantial.

**Metareview: Summary, Strengths And Weaknesses:**

This work develops a  pragmatic variant of the well-studied Correlation Clustering problem, with an emphasis on its “Minimizing Disagreements” version. We are given a budget on the number of queries allowed to make to an expensive oracle---perhaps a large ML model---that, given an edge e, returns the true labeling of e. We are also given unlimited access to a cheap but less-accurate such oracle, e.g., based on embedding-based models that can be queried fast. The paper develops extension of the KWIKCLUSTER algorithm that uses the cheaper oracle carefully to reduce the number of queries to the expensive oracle. They prove a 3-approximation for a stronger assumption on the cheap oracle, except for an additive error, and this additive error is shown to be inherent. The 3-approximation also runs faster than before. A thorough empirical analysis is performed to motivate the problem formulation, demonstrate the superior performance of their algorithm over baselines, and to conduct ablation studies on the tunable hyperparameters.

This framework is pragmatic, and helps one leverage weaker-yet-useful models for predicting similarities to obtain scalable algorithms. The theoretical results obtained match the results for the original version of the problem, except for the additive error. The algorithm is simple and efficient; the experiments support the claims of the paper. The paper is well-written.

The authors are encouraged to develop good bounds on the number of queries to the second oracle.


**Note From Pc:**

if the above contains the word "oral" or "spotlight" please see: "oral" presentation means -> notable-top-5% and "spotlight" means -> notable-top-25%. As stated in our emails, we are disassociating presentation type from AC recommendations